# On the functional brain networks involved in tool-related action understanding

Giovanni Federico [1✉], François Osiurak [2,3], Giuseppina Ciccarelli[1], Ciro Rosario Ilardi[1], Carlo Cavaliere[1], Liberatore Tramontano[1], Vincenzo Alfano[1], Miriana Migliaccio[1], Angelica Di Cecca[1], Marco Salvatore[1] & Maria Antonella Brandimonte[4]

Tool-use skills represent a significant cognitive leap in human evolution, playing a crucial role in the emergence of complex technologies. Yet, the neural mechanisms underlying such capabilities are still debated. Here we explore with fMRI the functional brain networks involved in tool-related action understanding. Participants viewed images depicting action-consistent (*e.g.*, nail-hammer) and action-inconsistent (*e.g.*, scarf-hammer) object-tool pairs, under three conditions: semantic (recognizing the tools previously seen in the pairs), mechanical (assessing the usability of the pairs), and control (looking at the pairs without explicit tasks). During the observation of the pairs, task-based left-brain functional connectivity differed within conditions. Compared to the control, both the semantic and mechanical conditions exhibited co-activations in dorsal (precuneus) and ventro-dorsal (inferior frontal gyrus) regions. However, the semantic condition recruited medial and posterior temporal areas, whereas the mechanical condition engaged inferior parietal and posterior temporal regions. Also, when distinguishing action-consistent from action-inconsistent pairs, an extensive frontotemporal neural circuit was activated. These findings support recent accounts that view tool-related action understanding as the combined product of semantic and mechanical knowledge. Furthermore, they emphasize how the left inferior parietal and anterior temporal lobes might be considered as hubs for the cross-modal integration of physical and conceptual knowledge, respectively.

[1] IRCCS SYNLAB SDN S.p.A., Naples, Italy. [2] Laboratoire d'Etude des Mécanismes Cognitifs (EA 3082), Université de Lyon, Bron, France. [3] Institut Universitaire de France, Paris, France. [4] Laboratory of Experimental Psychology, Suor Orsola Benincasa University, Naples, Italy. ✉email: research@giovannifederico.net

Tool use represents a human-characterizing ability and constitutes a critical component of the skills through which complex techniques and technologies may have emerged in our lineage. Indeed, one may easily situate tool use at the root of the radical transformation of the Earth's surface operated by *Homo sapiens* over the centuries[1–4]. Humans use the most diverse tools, from the simplest (*e.g.*, a hammer) to the most sophisticated (*e.g.*, a computer). In this study, we will focus on the first types of tools, namely prehensile objects that can extend the agent's sensorimotor potential[5]. Upon visual perception, these objects can activate action-related regions of the human brain, regardless of their immediate utility[6,7]. Nevertheless, such automatic activations upon seeing a tool do not necessarily reflect the observer's intention to utilize it, identify its function, or understand its potential in solving day-to-day problems[8–10]. These considerations form the epistemological core of the present study.

In everyday life, common interactions between humans and tools cannot be readily dissociated from the former's ability to recognize the latter's identity and functions. Thus, an individual may quickly recognize that a T-shaped object with a smooth wooden handle and a rectangular steel head is a hammer, belongs to the category of prehensile tools, and its prototypical function is to hammer[11]. However, such a kind of *semantic knowledge* about tools is not sufficient for individuals to use, for instance, a hammer on a nail: What is needed here is a "problem" to solve, even the simplest one, as, for example, hanging a picture on a wall in the living room. In fact, using the hammer to pound a nail into the wall constitutes an easy way to solve such a problem. The tool-use action is now goal-oriented and can be seen as an instance of a problem-solving situation[12,13]. At this stage, the "what," *i.e.*, semantic knowledge, gives way to the "how," namely *mechanical knowledge*, which enables the agent to understand how anchoring a nail in the wall may represent an excellent way to build a Newtonian system of forces that can prevent the picture from falling to the ground due to gravity[13–15]. Then, along with such implicit *technical reasoning* about the physical properties of the object-tool pair (*i.e.*, how to use the hammer with a nail), the agent should recruit *sensorimotor knowledge* (*i.e.*, the motor system) to grasp and use the hammer by making rhythmic oscillations with their arm[16]. The problem is solved, and the world has assumed a new goal-based physical configuration: The picture is on the wall.

What we briefly summarized above highlights how tool-related abilities in everyday life contexts require the interplay of multiple neurocognitive systems. Unsurprisingly, the debate about how the brain generates these skills is highly lively within the literature. From a neurocognitive side, such a scientific debate recognizes an essential point of synthesis in the macroscopic anatomical-functional differentiation of dorsal and ventro-dorsal brain regions, which are largely involved in action (*i.e.* the "how"), and ventral brain regions responsible for semantic knowledge (*i.e.*, the "what")[17–20]. From anthropological and evolutionary biology perspectives, the origins of complex tool-related action understanding can be traced back to the phylogenetic development of the posterior regions of the human brain, that is, the parietal lobes[21,22]. Compared to other mammals, primates underwent a notable expansion and functional specialization of the parietal lobes[2,23]. For instance, morphometric studies have demonstrated how the superior parietal lobe, particularly the precuneus, located on its medial wall, exhibits a significantly larger size in living humans compared to chimpanzees[24]. Larger parietal lobes in modern humans might be attributed to lengthened dorsal brain regions, including the precuneus and the Jensen sulcus, namely, a branch of the intraparietal sulcus located between the supra-marginal (SMG) and angular gyri[25,26]. The precuneus has been recently recognized as a crucial hub for visuospatial integration, namely, the coordination of different brain areas responsible for

visual processing, attention, and motor control[2,27,28]. Also, clinical neurology highlighted how damage to this brain region could result in visuospatial impairments[29,30]. Therefore, by looking at its functional specificity, evolutionary trajectory, and clinical findings, the precuneus may be assumed to be a compelling neural correlate for the advanced visuospatial abilities and, consequently, tool-use skills observed in humans.

Ventrally to the superior parietal lobes, another evolutionary critical region is the inferior parietal cortex (IPC). Indeed, comparative studies highlighted a prominent expansion of this region in primates[31]. The left IPC, particularly its area PF, which is located in the SMG, has been associated with the understanding of object-related mechanical actions (*i.e.*, the "how")[10]. Convergent evidence about the role of the left IPC in mechanical knowledge comes from clinical neurology, which highlighted how damage to this area may be associated with tool-use apraxia[13,32–34]. This clinical condition is characterized by the inability to reason about physical properties, resulting in difficulties with novel tool-use tasks and a decreased likelihood of employing trial-and-error strategies[35,36]. Along with the left area PF, moving anteriorly towards ventro-dorsal frontal regions, a second neural correlate of mechanical knowledge includes the left inferior frontal gyrus (IFG)[37–40]. Indeed, studies from different laboratories involving participants engaged in tool-use and action-understanding tasks have consistently found the involvement of the left area PF and left IFG in these tasks[17,38,41–43].

Alongside visuospatial and mechanical skills, semantic knowledge is involved in tool-related action understanding (*i.e.*, the "what")[44]. From an early age, we learn to navigate our surroundings, identify objects, and anticipate their behaviors[12,45]. With experience and learning, we understand physical reality through abstract and conceptual thinking, that is, tool-related semantic knowledge[46–48]. Moving more ventrally from the parietal and frontal regions described above, several studies have indicated how tool-related semantic knowledge involves extensive regions of the temporal cortex, mainly the left middle and inferior temporal lobe[17,18,32,49–51]. These brain regions play a critical role in multiple facets of tool-related action understanding. For instance, the middle temporal gyrus retrieves stored semantic knowledge about objects and tools. In contrast, the inferior temporal gyrus processes object-specific visual features and integrates tool-related semantic knowledge with other types of semantic knowledge located in more anterior temporal lobe regions[17,52–54]. From the clinical side, neurological findings have documented how patients with temporal lesions may exhibit intact motor skills to manipulate tools proficiently while experiencing a profound deficit in their ability to recognize the functions associated with these tools[55,56].

Above, we concisely described a sketch of a much more complex reality of neurocognitive systems involved in tool-related action understanding[32,40,41,57]. The reflection of this complexity is well testified by the substantial body of evidence accumulated over the past two decades, which elucidated the vast cognitive mechanisms involved in tool-related action understanding using different epistemological approaches and methodologies[40,58–60]. Behavioral studies align with neuroimaging findings, indicating that human understanding of tool-related actions arises from interactions among distinct and specialized neurocognitive systems. For instance, considering the well-established relationship between vision and cognition in humans[61], a series of studies documented how contextual and sensorimotor knowledge may directly affect the temporal allocation of visuospatial attention to different parts of the visual scene[62–65]. Complementary evidence highlighted the effects of semantic and higher-order knowledge in modulating how humans visually explore the environment and process its action possibilities[9,66–73].

Capitalizing on the research of the last two decades, most recent theoretical models assume tool-related action understanding as an integrated product of semantic, mechanical, and sensorimotor knowledge[9,19,72,74]. Notwithstanding, while the contribution of sensorimotor knowledge in tool-related action understanding has been extensively studied in the literature[75–79], the functional connectivity of tool-related semantic and mechanical brain systems do not enjoy the same level of investigation. Yet, the combined functioning of these brain systems in individuals' everyday life may give rise to flexible interpretations of physical reality. For instance, individuals may shift the functional knowledge of a screwdriver to a knife if they find themselves unscrewing a screw without having the screwdriver[80,81]. Such by-analogy problem-solving resolution strategies reflect the polymorphic nature of tool-related action knowledge. Hence, investigating the functional neural circuitry of semantic and mechanical knowledge may significantly advance our comprehension of tool-related action understanding.

This study used functional magnetic resonance imaging (fMRI) to explore dorsal, ventro-dorsal, and ventral functional brain systems involved in tool-related action understanding. We presented participants with three-dimensional computer-generated color images of object-tool pairs that were either semantically consistent (e.g., a screwdriver and a screw) or inconsistent (e.g., a screwdriver and a nut) for action. These stimuli were shown in three different experimental conditions: (i) a free-observation task (i.e., control condition), which required participants to view the object-tool pairs under natural/unconstrained condition; (ii) a looking-to-recognize task (i.e., semantic condition), which involved a yes-no recognition task where participants had to say whether a subsequently seen stimulus (i.e., a tool) was present or not in the previously viewed object-tool pair; (iii) a looking-to-use task (i.e., mechanical condition), where participants were asked to judge whether the tool of the object-tool pairs was usable with the object[9]. Then, we investigated the task-based functional connectivity[82,83] of the above-mentioned action-understanding-related brain regions (ROIs) while participants looked at the object-tool pairs. In particular, the analyses included as ROIs the precuneus, the left area PF of the inferior parietal cortex (IPC), the pars triangularis (IFG tri) and opercularis (IFG oper) of the inferior frontal gyrus. These ROIs represented the dorsal (precuneus) and ventro-dorsal (PF and IFG) brain systems[38,40]. Also, we considered the anterior (aMTG, aITG), posterior (pMTG, pITG), and temporo-occipital portions (toMTG, toITG) of the middle and inferior temporal gyrus as ROIs representing the ventral system[17,41,46]. We used the same object-tool pairs in all experimental conditions and limited the functional connectivity analysis to the duration of their presentation. By doing that, we aimed to exclude task- and motor-related responses that might affect brain activations. We hypothesized that functional left-brain co-activation patterns would differ within experimental conditions. Specifically, we expected that observing object-tool pairs in the semantic condition – when compared to the control condition – would yield a functional connectivity pattern involving wide ventral/temporal co-activations. On the other hand, when observing these same pairs in the mechanical condition and comparing it to the control condition, we predicted a co-activation pattern mainly involving dorsal and ventro-dorsal ROIs, including the area PF of the left inferior parietal cortex. Lastly, we posited that the temporal neural circuitry would play a role in distinguishing semantically consistent from semantically inconsistent pairs.

## Results

The functional connectivity analysis comprised ten left-brain ROIs (i.e., Precuneus, PF, IFG tri, IFG oper, aMTG, pMTG, toMTG, aITG, pITG, toITG; Fig. 1C). Pairwise ROI-to-ROI co-activations with a p-value < 0.05 corrected with False Discovery Rate (FDR)[84] are reported in Table 1 (semantic vs control condition), Table 2 (mechanical vs control condition), Table 3 (semantically consistent object-tool pairs), and Table 4 (semantically inconsistent object-tool pairs). The observation of semantically consistent object-tool pairs resulted in a coupling of co-activations predominantly localized in the left inferior- and middle-temporal regions (Table 3 and Fig. 1F). In contrast, looking at semantically inconsistent object-tool pairs co-activated the left inferior frontal gyrus and the medial and inferior temporal regions (Table 4 and Fig. 1G). Each experimental condition was contrasted with the control condition (i.e., semantic vs control condition and mechanical vs control condition). Results highlighted differences in the left fronto-temporo-parietal network we analyzed as the experimental condition varies. In particular, the semantic condition differed from the control condition for an interplay of functional connections between the precuneus and regions belonging to the ventro-dorsal (e.g., left IFG oper and left IFG tri) and ventral brain systems (e.g., left toMTG, left pMTG, left toITG, left pITG; Table 1 and Fig. 1D). Instead, the visual exploration of object-tool pairs in the mechanical condition, compared to the control condition, produced specific co-activation patterns in dorsal and ventro-dorsal brain regions, namely between left IFG tri and left PF, and between PC and left IFG tri and left IFG oper. Also, the left toITG appeared co-activated with PC and left IFG tri (Table 2 and Fig. 1E).

Results are summarized in Fig. 1D–G.

## Discussion

The present study used fMRI to explore the functional brain networks involved in tool-related action understanding. Participants were stimulated with computer-generated images depicting object-tool pairs that could be semantically consistent (e.g., notebook-pen) or inconsistent (e.g., bolt-knife) for action. These stimuli were presented in three experimental conditions: the semantic task (where participants determined the presence of a tool in previously viewed object-tool pairs), the mechanical task (involving the assessment of the usability of the tool with the paired object), and the control condition, in which participants simply observed the object-tool pairs without any specific tasks. Subsequently, we investigated task-based functional connectivity of specific left-brain regions associated with tool-related semantic and mechanical knowledge while participants looked at the object-tool pairs, within experimental conditions[17,38–41,46,85]. Importantly, we individually inspected the functional connectivity patterns as a function of the task type by comparing each experimental condition with the control condition, which served as the baseline. Also, we limited the analyses to the duration of object-tool pair presentation to prevent task- and motor-related responses that might affect brain co-activations. Results highlighted an interplay of functional co-activations in the context of the wide left fronto-temporo-parietal brain network we considered (i.e., Precuneus, PF, IFG tri, IFG oper, aMTG, pMTG, toMTG, aITG, pITG, toITG), which varied as a function of the experimental manipulations.

At the first level of analysis, namely, irrespective of the experimental condition, semantically consistent object-tool pairs co-activated regions localized in the left inferior- and middle-temporal regions (i.e., left aITG, pITG, pMTG, toMTG), while semantically inconsistent pairs co-activated left frontal regions (i.e., IFG) in addition to left medial and inferior temporal areas (i.e., aITG, pITG, aMTG, pMTG, toMTG). Such a temporal-centered neural circuitry seems to reflect the predictions of most

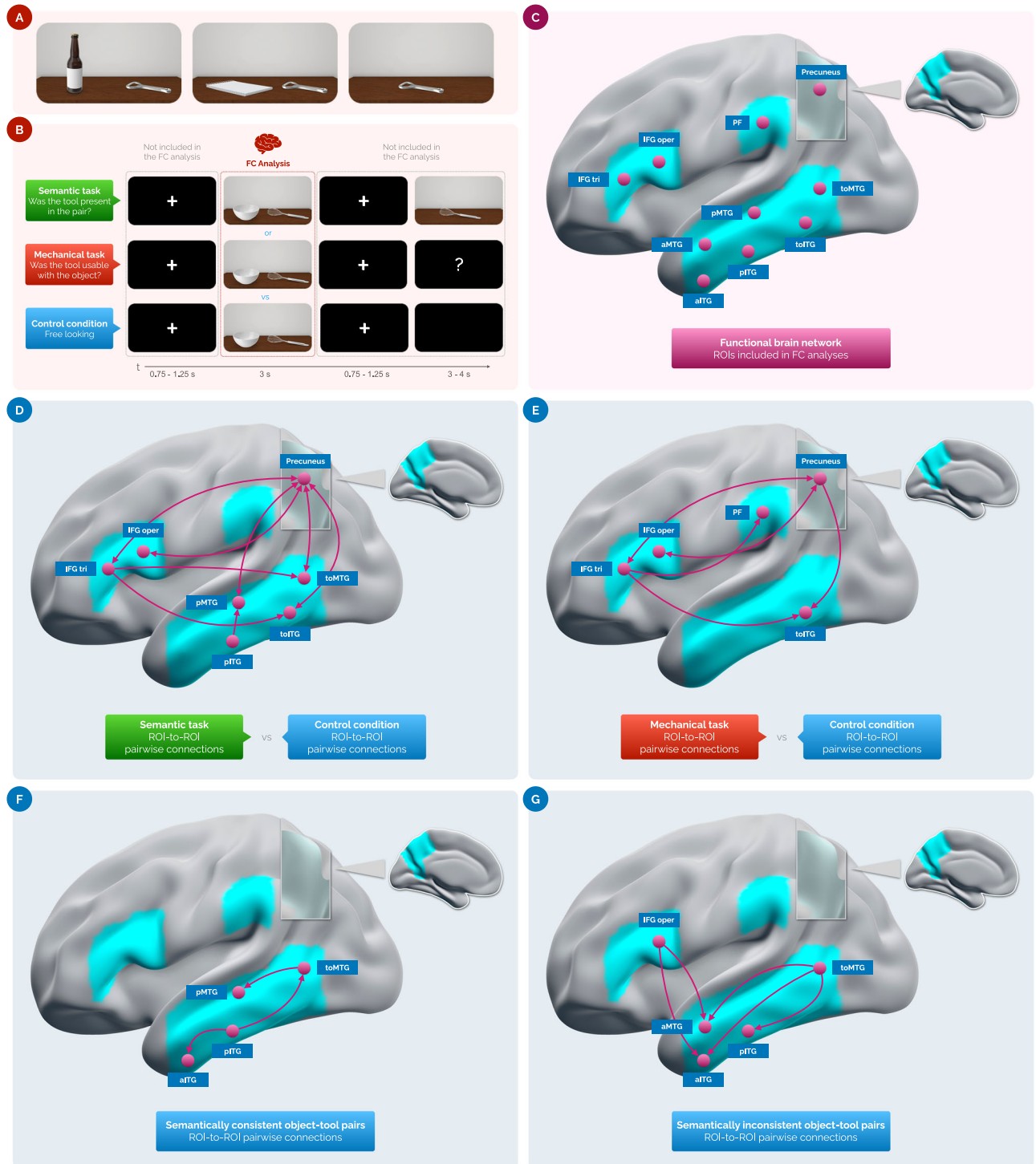

recent models of semantic cognition, which highlights how the temporal lobe may serve as a hub for the cross-modal representation of semantic knowledge by bringing together various sources of information[46,48]. Notably, the encoding of stimuli in the temporal cortex is inconsistent throughout its different subregions. The superior temporal gyrus responds more strongly to auditory stimuli, while the inferior temporal gyrus (ITG) is more inclined towards visual inputs[86]. Accordingly, we found that ITG was co-activated along with the middle temporal gyrus (MTG), a temporal region that responds to multiple input modalities and is believed to play a key role in the multimodal integration of semantic knowledge[86].

We used a specific kind of visual stimuli, namely object-tool pairs, to manipulate the semantic consistency prompted by the visual scene. Precisely, we manipulated the functional relationship between the items of the object-tool pairs. In human beings, semantic knowledge comprises multiple interconnected pieces of information whose most straightforward modes of organization are "taxonomic" and "thematic" relations[87,88]. Taxonomic relations are hierarchical and based on shared features, while thematic relations are based on extrinsic relations between objects, such as functional relations. Consequently, relations between objects and tools of the object-tool pairs we used in this study can be considered thematic relations. Results highlighted left

**Fig. 1 Experimental stimuli, design, and results. A** Example of stimuli involved in the study. From left to right: a semantically consistent object-tool pair (*i.e.*, a bottle and a bottle opener), a semantically inconsistent object-tool pair (*i.e.*, a notebook and a bottle opener), and a single tool (*i.e.*, a bottle opener). **B** The three experimental conditions. From top to bottom: (i) the semantic task, in green; (ii) the mechanical task, in light red; (iii) the control condition, in blue. Functional connectivity (FC) analyses included only the duration when the object-tool pairs were presented. The object-tool pairs were the same in all experimental conditions. **C** The ten ROIs of the left human brain we included in FC analyses. **D** The contrast between semantic condition and control condition revealed FC differences between the PC and regions belonging to the ventro-dorsal brain systems, such as the left IFG *oper* and left IFG *tri*, as well as the ventral brain system, including the left toMTG, left pMTG, left toITG, and left pITG. See Table 1 for ROI-to-ROI FC values and statistics. **E** Compared to the control condition, the mechanical condition resulted in specific FC patterns in the ventro-dorsal brain areas, particularly between the left IFG *tri* and left PF, and between the PC and left IFG *tri* and left IFG *oper*. The ventral region left toITG appeared co-activated with the PC and left IFG *tri*. See Table 2 for ROI-to-ROI FC values and statistics. **F** Observing semantically consistent object-tool pairs resulted in co-activations localized in the left inferior and middle temporal regions. See Table 3 for ROI-to-ROI FC values and statistics. **G** Semantically inconsistent object-tool pairs co-activated the left inferior frontal gyrus and the left medial and inferior temporal areas. See Table 4 for ROI-to-ROI FC values and statistics. In **D**–**G** the violet lines indicate task-based ROI-to-ROI functional connectivity. In **C**–**G** the left fronto-temporo-parietal functional brain network considered in this study is highlighted in light blue, with violet dots indicating the ROIs included in the FC analyses. All the 3D sagittal brain meshes were generated using Surf Ice (v. 1.0.20211006; https://www.nitrc.org/projects/surfice). PF area PF of the left inferior parietal cortex, IFG *tri pars triangularis* of the left inferior frontal gyrus, IFG *oper pars opercularis* of the left inferior frontal gyrus, toITG temporo-occipital part of the left inferior temporal gyrus, pITG posterior part of the left inferior temporal gyrus, aITG anterior part of the left inferior temporal gyrus, toMTG temporo-occipital part of the left middle temporal gyrus, pMTG posterior part of the left middle temporal gyrus, aMTG anterior part of the left middle temporal gyrus, FC functional connectivity, ROI(s) region(s) of interest, 3D three-dimensional.

---

**Table 1 FC Analysis – ROI-to-ROI pairwise connections – Semantic vs Control condition.**

| Seed | Target | Beta | T(18) | P |
|------|--------|------|-------|---|
| PC | toMTG | 0.36 | 5.27 | 0.001 |
| PC | IFG *oper* | 0.50 | 4.77 | 0.001 |
| PC | IFG *tri* | 0.49 | 4.14 | 0.002 |
| PC | toITG | 0.56 | 4.05 | 0.002 |
| IFG *oper* | PC | 0.28 | 4.46 | 0.003 |
| IFG *tri* | PC | 0.23 | 4.26 | 0.004 |
| toMTG | PC | 0.20 | 4.17 | 0.005 |
| toITG | PC | 0.16 | 4.05 | 0.007 |
| PC | pMTG | 0.30 | 2.92 | 0.017 |
| IFG *tri* | toITG | -0.16 | -2.71 | 0.041 |
| IFG *tri* | toMTG | -0.18 | -2.59 | 0.041 |
| pITG | pMTG | -0.18 | -3.22 | 0.042 |
| pMTG | PC | 0.16 | 3.18 | 0.047 |

*PC* Precuneus, *IFG tri pars triangularis* of the left inferior frontal gyrus, *IFG oper pars opercularis* of the left inferior frontal gyrus, *toITG* temporo-occipital part of the left inferior temporal gyrus, *pITG* posterior part of the left inferior temporal gyrus, *toMTG* temporo-occipital part of the left middle temporal gyrus, *pMTG* posterior part of the left middle temporal gyrus. *P* p-value with False Discovery Rate correction.

**Table 2 FC Analysis – ROI-to-ROI pairwise connections – Mechanical vs Control condition.**

| Seed | Target | Beta | T(18) | *P* |
|------|--------|------|-------|-----|
| PC | IFG *tri* | 0.36 | 3.35 | 0.018 |
| PC | IFG *oper* | 0.42 | 3.29 | 0.018 |
| IFG *tri* | PF | -0.15 | -3.20 | 0.023 |
| IFG *tri* | PC | 0.18 | 3.18 | 0.023 |
| IFG *tri* | toITG | -0.17 | -2.86 | 0.031 |
| PC | toITG | 0.34 | 2.79 | 0.036 |
| IFG *oper* | PC | 0.26 | 3.29 | 0.037 |

*PC* Precuneus, *PF* area PF of the left inferior parietal cortex, *IFG tri pars triangularis* of the left inferior frontal gyrus, *IFG oper pars opercularis* of the left inferior frontal gyrus, *toITG* temporo-occipital part of the left inferior temporal gyrus. *P* p-value with False Discovery Rate correction.

**Table 3 FC Analysis – ROI-to-ROI pairwise connections – Semantically consistent object-tool pairs.**

| Seed | Target | Beta | T(18) | *P* |
|------|--------|------|-------|-----|
| toMTG | pMTG | -0.89 | -3.19 | 0.046 |
| pITG | toMTG | -0.51 | -3.27 | 0.038 |
| pITG | aITG | -0.75 | -2.86 | 0.047 |

*pITG* posterior part of the left inferior temporal gyrus, *aITG* anterior part of the left inferior temporal gyrus, *toMTG* temporo-occipital part of the left middle temporal gyrus, *pMTG* posterior part of the left middle temporal gyrus. *P* p-value with False Discovery Rate correction.

**Table 4 FC Analysis – ROI-to-ROI pairwise connections – Semantically inconsistent object-tool pairs.**

| Seed | Target | Beta | T(18) | *P* |
|------|--------|------|-------|-----|
| IFG *oper* | aITG | -1.56 | -3.57 | 0.014 |
| IFG *oper* | aMTG | -0.93 | -3.42 | 0.014 |
| toMTG | pITG | -0.87 | -3.50 | 0.023 |
| toMTG | aITG | -1.69 | -2.99 | 0.024 |
| toMTG | aMTG | -1.09 | -2.98 | 0.024 |

*IFG oper pars opercularis* of the left inferior frontal gyrus, *pITG* posterior part of the left inferior temporal gyrus, *aITG* anterior part of the left inferior temporal gyrus, *toMTG* temporo-occipital part of the left middle temporal gyrus, *aMTG* anterior part of the left middle temporal gyrus, *P* p-value with False Discovery Rate correction.

---

associations between actions and meanings[89–91]. Notably, these regions are part of a semantic control network whose activation is maximized during tasks involving weak associations among objects or tasks involving competition or ambiguity among stimuli[92–97]. Thus, when there is less semantic consistency across stimuli or tasks require information that is not the strongest in memory, additional processing may be needed to enable semantic retrieval. Convergent clinical evidence shows how stroke-aphasia patients may display impaired controlled semantic retrieval due to damage to these regions[55,98].

Globally, the overlap in the left temporal and frontal regions we reported is particularly interesting when viewed through the lens of the so-called *Hub-and-Spoke* hypothesis of semantic knowledge[46,48]. This hypothesis attempts to connect the gap between modal and a-modal theories of conceptual knowledge, proposing how semantic knowledge may be formed by

frontotemporal functional co-activations, including the posterior part of the left MTG (pMTG) for thematically consistent pairs (*e.g.*, nail-hammer) and the left IFG for thematically inconsistent pairs (*e.g.*, nut-screwdriver). Both left IFG and pMTG support semantic processing of thematic knowledge by handling complex

interactions between modality-specific sources of information, namely *spokes* (*e.g.*, visual features, somatosensory information, praxis, and sounds), and a trans-modal semantic *hub* that provides a modality-independent representational resource, namely, the anterior temporal lobe (ATL). The interplay between the *spokes* and the *hub* results in consistent and generalizable concepts. Hence, this study's findings align with the literature underlining the role of the temporal lobe in high-level semantic processing and extend it to the epistemological domain of object- and tool-related action understanding[9,17,66,67,99,100].

At the second level of investigation, we aimed to explore the functional connectivity patterns involved in tool-related semantic and mechanical knowledge. Results indicated that the semantic condition produced extensive ventralized functional co-activations. Additionally, we observed dorsal and ventro-dorsal co-activations that involved the precuneus and left IFG. The ventral co-activations were centered around left temporo-medial brain structures, including the left pMTG, toMTG, and toITG. In the context of an explicit semantic task like the one we used in this study, *i.e.*, the yes-no recognition task, co-activations of such temporal regions appear to be in line with the predictions of the semantic cognition models described above as well as with the most recent evidence that has shown the influence of top-down semantic processing in tool-related visual cognition[9,41,46,65–67,69,99,101]. In such a theoretical framework, research has shown how the left temporal co-activations we found may be involved in different aspects of tool-related semantic processing, including the retrieval of stored semantic knowledge about objects and tools (toMTG), the processing of object-specific visual features (ITG), and the integration of semantic information about objects and tools with other kinds of semantic knowledge (pMTG)[17,52–54]. Reasonably, the explicit goal of the recognition task may have led participants to semantically process the visual scene right from the earliest stages of visual encoding, irrespective of the action readiness prompted by the visual-perceptual context. Therefore, this result may offer a neurocognitive reality to most recent behavioral findings that emphasize the pre-eminence of semantic over sensorimotor processing in tool-related action understanding[11,65,99]. More generally, these results stress that semantic knowledge might be essential to create generalizable representations about tools and objects[85,102].

Shifting the focus to the mechanical condition in which participants were asked to judge the usability of the tool on the object immediately after seeing the object-tool pairs, we found a pattern of functional ventro-dorsal co-activations involving the area PF of the left inferior parietal lobe[103]. This result aligns with the most recent evidence highlighting the role of such an area in technical reasoning[17,38–40]. Technical reasoning can be described as the ability to reason about the physical properties of objects, such as solidity, rigidity, and weight[5,104]. It belongs to the conception level, and it is characterized by causal and analogical reasoning, which involves predicting the effects of actions on the environment and transferring knowledge from one situation to another[10,105,106]. This reasoning is based on mechanical knowledge, which is acquired through experience and refers to non-declarative knowledge about physical principles such as cutting or lever actions. The result of technical reasoning is a mental simulation of the mechanical actions required to perform a task (*e.g.*, hammering), which aids in guiding the activity of the motor-control system[16]. Thus, when a mechanical action is conceived (*e.g.*, driving a nail into a wall with a hammer), it constrains the motor system in planning, selecting, and controlling motor programs. If a problem occurs (*e.g.*, the nail cannot be driven into the wall), new mechanical knowledge must be recruited (*e.g.*, increasing the percussion power by changing how the hammer is grasped). This will constrain the motor system in selecting,

planning, and controlling new motor programs. Nevertheless, the motor system itself may impose constraints. For instance, when someone comes up with the idea that a hammer with a heavy head would help drive a nail into the wall, difficulties in manipulating the hammer because of its weight may lead to generating new mechanical solutions. In other words, the interaction between technical reasoning and the motor-control system is consistent with the ideomotor principle[107].

The technical reasoning hypothesis was initially developed in clinical neuropsychology by studying left-brain-damaged patients with apraxia of tool use[13,32,49,108–110]. These patients struggle with selecting and using familiar tools, such as a hammer or a knife, and find it challenging to choose, utilize, or make novel tools to solve mechanical problems. In these patients, tool-use disorders might be due to their inability to reason about physical object properties, as they show difficulty in novel tool-use tasks and are less likely to adopt trial-and-error strategies[36]. Multiple studies have shown that tool-use disorders are more likely to occur after damage to the left IPC, particularly the left area PF within the supramarginal gyrus[32–34]. A recent meta-analysis of neuroimaging studies has confirmed that the left area PF is involved in processing mechanical actions rather than motor actions and is activated when people focus on mechanical actions[40]. Also, observing others performing mechanical actions engages the same network, and the left area PF is preferentially activated when people observe tool-use actions. Therefore, people may reason technically even when observing others perform mechanical actions[110,111]. Finally, a recent morphometric study showed that the cortical thickness (*i.e.*, a volume-related measure associated with cognitive performance as reflecting the size, density, and arrangement of cells in a brain region) of the left area PF mediates technical-reasoning skills[38].

Along with the left area PF, the left IFG is also involved in technical reasoning. This brain region has been consistently implicated in everyday tool-use skills[41,112]. The finding of left IFG co-activations in this study aligns with existing literature. In broad terms, the IFG is part of the brain networks associated with executive functions, involving cognitive control processes like task switching, attentional selection, conflict monitoring, and integration of information from multiple sources[92,113–115]. Therefore, the IFG might play a key role in integrating action-related information of diverse nature. This perspective aligns with this study's results, as the IFG consistently showed co-activation across all experimental conditions. The left inferior prefrontal cortex may play a critical role in tool-related action understanding as this brain region has been widely implicated in motor timing, sequencing, and simulation[116–118]. Broadly speaking, the involvement of the frontal lobe in tool-related action understanding may suggest the presence of inhibitory/attentional mechanisms that regulate the selection of the most appropriate actions from those prompted by the visual-perceptual context[119]. Additionally, specific visual-attentional effects may have affected the interplay of the left frontoparietal regions we observed in this study. For instance, it has been found that eye movement patterns significantly influence brain activity in these regions[79]. However, the involvement of frontal areas in tool use has not been extensively explored in the literature. This may be due to the lack of a clear correlation between tool-use impairments and frontal lesions (but see[120–122]). Nonetheless, as demonstrated by this study, frontal regions may actively contribute to tool-related action understanding and should be extensively investigated in future investigations.

Not only the IFG but also the precuneus exhibited co-activations in all the experimental conditions. Such a finding is particularly suggestive when considering the evolutionary trajectory of this brain region in modern humans[1,2,22,27,28]. Indeed,

most recent research highlighted how the precuneus underwent significant growth during human evolution. Studies involving endocasts and shape analysis to examine the brain's structure[1,123] have found that the braincase of modern humans is more globular and that the parietal region bulges more than in subspecies of archaic humans, such as Neanderthals[22]. The significant expansion of the precuneus throughout the evolutionary history of *Homo genus* has been linked to advanced tool-use abilities and the creation of complex objects and artifacts essential for activities such as foraging, defense, housing, and other daily tasks performed by individuals and communities[2]. Additionally, along with the precuneus, both the superior and inferior parietal lobes have seen an increase in cortical complexity, and this may support behaviors unique to humans, such as technological innovation, self-awareness, numerosity, mathematical reasoning, and language, namely, functions crucial to material culture[2,22,124]. Thus, on closer looking, two distinct lines of evidence seem to coexist in the literature. The neurocognitive literature highlights the significant role of tool use and action-understanding abilities for humans, while the anthropological literature emphasizes the increasing complexity of tools, techniques, and technologies within our lineage across generations[125,126]. The parietal lobes act as critical points of convergence for tool use and technology evolution, thereby providing an epistemological bridge between the neuroscientific and anthropological literature to study such fascinating and human-characterizing abilities[124].

The results of this study fit well with the most recent integrated models that stress the role of interactions among neurocognitive systems in action understanding. In particular, the "action reappraisal" construal has been recently introduced in the literature as a neurocognitive mechanism enabling agents to make sense of reality by integrating distinct kinds of action-related information[72]. On the one hand, the action reappraisal construal implies that individuals may "grasp" action possibilities based on their goals and needs. On the other hand, such an action propensity may be influenced by environmental contexts, prior knowledge, and individual experience. By extending the original Gibsonian notion of "affordances"[127], the action reappraisal construal highlights how action understanding may rely on dynamic and consistent-with-agent-intentions interactions involving neurocognitive systems associated with semantic and mechanical knowledge. Interestingly, the distinction between semantic (*i.e.*, the "what") and mechanical understanding (*i.e.*, the "how") mirrors the difference stressed in both the archaeological and anthropological literature between *knowledge* and *know-how*[128–131]. Most importantly, the dissociation between semantic and mechanical knowledge is supported by clinical neurology. Patients with semantic dementia, *i.e.*, a rare form of fronto-temporal dementia, can manipulate tools but cannot name them or recognize their functions. In contrast, patients with limb apraxia, *i.e.*, a clinical condition resulting from left parietal lesions, can recognize tools but cannot manipulate them appropriately[13,49,55,56]. While clinical and anthropological findings highlight dissociations between semantic and mechanical knowledge, in healthy individuals, such neurocognitive systems may interact to provide a framework through which they may understand the physical world[99]. This study provides initial support for such an integrated interpretation of action understanding. Subsequent investigations with larger samples and data-driven approaches should further explore these aspects.

In conclusion, this study may help consolidate some hypotheses that are gradually emerging in the literature. These hypotheses view, on the one hand, the left inferior parietal lobe—particularly its area PF—as a key brain region for generating mechanical knowledge[38,39,42,43,124,132]. On the other hand, the temporal lobe—especially its anterior part—is viewed as the brain

seat of trans-modal integration of semantic knowledge[41,43,49,81,82,85]. At the same time, this study corroborates the role of the precuneus in action-related visuospatial skills[2,19,22,24,25] and lends support to recent accounts highlighting the involvement of the left inferior frontal gyrus in multi-modal information integration[133,134]. These findings suggest an integrated perspective on tool-related action understanding incorporated into the human brain's physical (dorsal and ventro-dorsal brain systems) and conceptual (ventral brain systems) components[9,72,74]. At a theoretical level, by speaking on the way the brain may deal with tools in everyday-life contexts, this study may foster the debate about how humans may *understand* and *manipulate* the environment to devise and use complex tools, techniques, and technologies[125].

## Methods

**Participants**. Twenty right-handed participants (11 females; mean age = 25.7 years, SD = 3.87) were enrolled in this study according to the following criteria: (i) lack of current or past history of alcohol/drug abuse; (ii) lack of current or past history of psychiatric illnesses; (iii) lack of history of brain injury, stroke, or any other significant clinical condition; (iv) lack of current or past use of psychoactive medications. The present study employed Python scripts based on *neurodesign* to determine the optimal design and sample size to increase the generalizability of the findings and enhance the study's reliability and validity[135]. The sample size for this study is consistent with similar studies[42,43]. Before starting the experiment, an expert medical doctor assessed participants through a clinical interview. The Edinburgh Handedness Inventory assessed participants' handedness[136]. One female participant was excluded from the analyses due to the onset of a panic attack a few minutes before finishing the MRI session. All participants gave written informed consent to their participation in the study. Participants did not receive any financial compensation for their participation in this study.

**Materials**. Thirty three-dimensional (3D) computer-generated images were divided into three different groups of stimuli. The first group of stimuli was composed of ten 3D color images depicting semantically consistent object-tool pairs (*i.e.*, the object on the left, *e.g.*, a glass, and the tool on the right, *e.g.*, a bottle). This group included the following stimuli: nail-hammer, bowl-kitchen whisk, carton box-cutter, bottle-bottle opener, screw-screwdriver, salami-knife, coffee cup-teaspoon, notebook-pen, glass-bottle, padlock-key. The second class of stimuli was composed of 3D color images depicting semantically inconsistent object-tool pairs (*i.e.*, the object on the left, *e.g.*, a bolt, and the tool on the right, *e.g.*, a knife). This group comprised the following object-tool pairs: scarf-hammer, women shoe-kitchen whisk, alarm clock-cutter, notebook-bottle opener, nut-screwdriver, bolt-knife, Christmas ball-teaspoon, men shoe-pen, cap-bottle, baseball-key. The third class of stimuli was composed of ten 3D color images depicting the same single tools as included in the object-tool pairs, namely a hammer, a kitchen whisk, a cutter, a bottle opener, a screwdriver, a knife, a teaspoon, a pen, a bottle, and a key. For the semantic condition (see *Experimental tasks*), we used all the stimuli (*i.e.*, object-tool pairs and single tools). Instead, we used only the stimuli depicting object-tool pairs for mechanical and control conditions (see *Experimental tasks*). All the objects and object-tool pairs depicted in the stimuli were graphically represented on a table in such a way as to appear closest to the observer, namely in the participant's peri-personal space. By considering the horizontal line of the table as a reference, object-tool pairs were placed at a mean perceived object-to-tool distance of approximately 20 cm and a center-to-center angle of vision of

approximately 180 deg. Information regarding distance and the angle of vision was derived from the estimations provided by the participants, as reported during the post-experimental interview. Stimuli were realized with Adobe Photoshop version 21.0.6 (Adobe, Inc., San Jose, CA, US) by including vectorial files of the PixelSquid library (TurboSquid, Inc., New Orleans, US; https://www.pixelsquid.com). The stimuli were depicted in an MRI-compatible binocular visor with a resolution of 1024x768px and a refresh rate of 60 Hz. The stimuli were selected by jury evaluation. The jury comprised ten individuals (5 females; mean age = 26.4 years; SD = 2.1). The jury members were distinct from the participants in the fMRI study. We asked the jury to rate each object from the study's stimuli on 2 Likert scales ranging from 1 (extremely difficult to grasp and use, extremely unrealistic) to 5 (extremely easy to grasp and use, extremely realistic). The stimuli received high scores in terms of ease of grasp (M = 4.6; SD = 0.52) and realism (M = 4.8; SD = 0.42). Examples of stimuli used in this study are summarized in Fig. 1A.

**Experimental tasks**. Three experimental conditions were conceived and implemented as experimental tasks to manipulate tool-related semantic/mechanical knowledge. Specifically, the following three tasks were devised: (i) a yes-no recognition task (*i.e.*, the semantic condition); (ii) a yes-no looking-to-use task (*i.e.*, the mechanical condition); (iii) a free-observation task (*i.e.*, the control condition). In the yes-no recognition task, participants were engaged in a slightly modified and MRI-adapted version of the short-term recognition task originally devised by Federico and Brandimonte[9]. Participants had to indicate in the faster time possible whether the latter stimulus (*i.e.*, a single tool) was present in the previously observed object-tool pair. In the yes-no looking-to-use task, participants had to indicate whether the tool (*e.g.*, a hammer) was usable on the object (*e.g.*, a nail or a scarf). The free-observation task acted as the control condition, requiring participants to look at the object-tool pairs most naturally (*i.e.*, no explicit task). The tasks involved in this study and the experimental visual flow are summarized in Fig. 1B.

**Procedure**. This study was conducted at the *Istituto di Ricovero e Cura a Carattere Scientifico* (IRCCS) Synlab SDN (Naples, Italy). Participants were randomly recruited through advertisements posted on social media networks. All ethical regulations relevant to human research participants were followed. Therefore, the study received approval from the local Ethics Committee (approval number 8/20 issued by Istituto Nazionale Tumori IRCCS "Fondazione G. Pascale", Naples, Italy). Before entering the scanner, participants familiarized themselves with the study's tasks using a dedicated personal computer in an experimental room other than the one containing the MR scanner. To avoid learning effects, the stimuli used in the practice tasks did not re-appear in the experimental tasks. As soon as participants were inserted into the scanner, T1-weighted images were acquired, and then a functional protocol was initiated (see *MRI data acquisition*). A within-subject hybrid experimental design was implemented. Specifically, the three experimental tasks were run consecutively, interspersed with a two-minute pause in which participants observed a black screen. Each run lasted about six minutes and was preceded by a text cue informing participants about the task. Both runs and trials were fully randomized across participants. To avoid confounding effects of motor execution on brain activations in semantic and mechanical tasks (*i.e.*, tasks where a yes-no response was required), participants were asked to respond only to a specific response trial, which was subsequently excluded from the fMRI analyses (see the experimental visual flow in Fig. 1B). Participants indicated their yes-no responses by

pressing one of two dummy-pad buttons using the index or the middle finger of their left non-dominant hand. The participants' responses were not recorded because the dummy pad used during the experiment was non-functional and served only as a tool for manipulating the experimental conditions.

**MRI data acquisition**. Participants' structural and functional MR images were acquired with a Philips Achieva dStream 3 T scanner and a 32-channel head coil. Blood-Oxygen Level Dependent (BOLD) images were recorded with T2-weighted Echo-Planar Images (EPI) acquired with the multi-band sequence. Functional images were collected as oblique-axial scans aligned with the anterior commissure–posterior commissure (AC–PC) line with the following parameters: 162 volumes per run, 45 slices, TR/TE = 2000/21.4 ms, flip angle = 90°, field of view = $240 \times 240$ mm$^2$, slice thickness = 3 mm, voxel size = $3 \times 3 \times 3$ mm$^3$, multiband factor = 2. Structural T1-weighted images were collected using a 3D T1-TFE sequence (180 sagittal slices, TR/TE = 8.1/3.7 ms, flip angle = 8°, field of view $240 \times 240$ mm$^2$, slice thickness = 1 mm, voxel size = $1 \times 1 \times 1$ mm$^3$).

**MRI data pre-processing and denoising**. MRI data were converted to Brain Imaging Data Structure (BIDS) by devising custom-made Python scripts. MRI data were visually inspected by an experienced neuroradiologist (C.C.) for quality check. MRI data pre-processing and denoising were performed using the Functional Connectivity Toolbox version 21a (CONN; https://www.nitrc.org/projects/conn) implemented in MATLAB version R2021b (The MathWorks, Inc., Natick, MA, US). Pre-processing was carried out by implementing the standard CONN pre-processing pipeline, which included the following steps: (i) functional realignment and unwarp; (ii) slice-timing correction; (iii) outlier identification with ART-based scrubbing; (iv) direct segmentation and normalization in the Montreal Neurological Institute (MNI) reference space; (v) 8-mm full-width at half-maximum (FWHM) Gaussian smoothing. Then, pre-processed data were denoised using the CONN's default linear detrending, de-spiking and filtering (*i.e.*, 0.008 Hz < $f$ < 0.09 Hz).

**Functional connectivity analysis**. Task-dependent changes in functional connectivity (FC) were investigated by devising a generalized form of context-dependent psychophysiological interactions (gPPI) with CONN[82]. gPPI is an implementation of psychophysiological interactions (PPI) that estimates effective FC for more than two experimental conditions. gPPI-based general linear models include: (i) the psychological predictors, *i.e.*, the task effects convolved with a canonical hemodynamic response function; (ii) the physiological predictors, *i.e.*, the time series associated with the brain regions of interest (ROIs); (iii) the interaction between the psychological and physiological predictors[82,137]. A hypothesis-driven gPPI analysis was performed to identify task-modulated changes in FC patterns covarying with the experimental conditions in the context of this study's hybrid experimental design. For each trial of each run, the psychological predictors of the FC analysis were modelled by considering the appearance on screen of the object-tool stimulus from its onset to its end (see the experimental visual flow in Fig. 1B). In this way, the FC analysis was restricted to the participants' visual encoding of the object-tool pairs, mitigating the confounding effects of the response trials. Multiple cortical areas of the left hemisphere were selected as ROIs from the CONN's default cortical atlas, *i.e.*, the Harvard-Oxford atlas as distributed with FSL (https://fsl.fmrib.ox.ac.uk). The ROI of the area PF of the left inferior parietal cortex was generated with CONN and included in the analyses. This ROI consisted of a 5-mm spheric kernel centered on the following MNI

coordinates: x = -55, y = -32, z = 35. The FC analysis included the following ten ROIs: the left area PF; the *pars triangularis* (IFG *tri*) and *pars opercularis* (IFG *oper*) of the left Inferior Frontal Gyrus; the anterior (aMTG), posterior (pMTG) and temporo-occipital (toMTG) part of the left Middle Temporal Gyrus; the anterior (aITG), posterior (pITG) and temporo-occipital (toITG) part of the left Inferior Temporal Gyrus; the Precuneous (PC). FC analyses were devised by positing the existence of a shared source of variance among the experimental conditions—specifically, sensorimotor knowledge. This knowledge is linked to the activation of sensorimotor brain regions when observing tools, regardless of the task demands[6,7]. Secondly, we hypothesized the existence of two distinct functional networks: the tool-related semantic network and the mechanical network[40,44], represented by the ROIs included in the analyses (see above). Accordingly, differences in FC between experimental conditions were analyzed by contrasting the two experimental tasks with the control task, which acted as the baseline. This approach reduced the variance associated with sensorimotor knowledge across all experimental conditions, facilitating the individual investigation of brain networks related to semantic and mechanical knowledge. Also, sensorimotor-related ROIs[6,7] were excluded from the analyses to maintain consistency with the study's experimental hypotheses. A second ROI-to-ROI analysis investigated FC differences as a function of object-tool pairs' semantic consistency, irrespective of the experimental condition. A p-value = 0.05 corrected by False Discovery Rate (FDR)[84] was used as a significance threshold for pairwise ROI-to-ROI connections.

**Reporting summary**. Further information on research design is available in the Nature Portfolio Reporting Summary linked to this article.

## Data availability
The data supporting the present study's findings are available from the corresponding author upon reasonable request.

## Code availability
The code supporting the present study's findings are available from the corresponding author upon reasonable request.

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

## Acknowledgements

We thank Prof. Lewis Wheaton and the other two anonymous reviewers for their valuable contributions and constructive feedback. Ricerca Corrente's grant from the Italian Ministry of Health supported this work. The funders had no role in study design, data collection and analysis, publication decisions, or manuscript preparation.

## Author contributions

G.F. and M.A.B. conceived the study. G.F. designed the study and the software experimental paradigm. G.F., V.A., and L.T. performed the experiment. C.C., V.A., and L.T. supervised magnetic resonance imaging (MRI) procedures and performed MRI quality checks. G.F. analyzed the MRI data and performed statistical analyses. G.C. and C.R.I. provided support for data analyses. M.S. acquired the financial support for the project leading to this study. G.F. was the main contributor to the writing of the manuscript and wrote its first draft. C.C., C.R.I., F.O., and M.A.B. revised drafts providing critical theoretical arguments. A.D.C. and M.M. provided support in data curation and formatting and typesetting the manuscript, tables, and figures. All authors approved the final manuscript.

## Competing interests

The authors declare no competing interests.
