## [Peer Review File · Communications Biology]

Reviewers' comments:

Reviewer #1 (Remarks to the Author):

For this review, I will use the recommended review framework provided by the journal in answering 5 core questions. As well, I am referencing several of my papers for consideration. This is not an attempt to collect citations, but an attempt to think about the results in a broader context, which I believe would be helpful.

Lewis Wheaton

1. What are the major claims of the paper?

This manuscript focuses on a very compelling research interest related to the understanding of the neural basis for our understanding of tool usage. This has been a long-standing issue of the field. The study results "considers action understanding as an ability emerging from semantical/conceptual and mechanical/sensorimotor knowledge integration".

2. Are they novel and will they be of interest to others in the community and the wider field?

I am not very convinced that these conclusions are entirely novel. There has been a longstanding argument that conceptual and sensorimotor knowledge are integrated in action understanding, which is a presumption of many recent imaging studies. It is novel to provide FC analysis as the primary approach, but much of the discussion focuses on "activations" or "co-activations" that are not included in this paper (unless I am missing some other figures).

3. If the conclusions are not original, it would be helpful if you could provide relevant references.

The authors present some of their work focusing on semantic understanding of tool use, but the authors have not included some science that presents both complimentary and diverging arguments to theirs. These include many papers from my own lab, including:

A behavioral study indicating how action may emerge through visual context

Borghi, A.M., Flumini, A., Natraj, N., Wheaton, L.A. One hand two objects: emergence of affordance in context. *Brain and Cognition*; 2012; 10: 64-73.

Gaze tracking studies that present a slightly different picture of visual attention driven by action context - Natraj N, Pella YM, Borghi AM, Wheaton LA. Visual encoding of tool-object affordances. *Neuroscience*, 2015, 310: 512-527.

Gaze + EEG demonstrating how visual attention modulates left and right hemisphere activity during tool use perception

- Natraj N, Alterman BM, Basunia S, Wheaton LA. The role of attention and saccades on parietofrontal encoding of contextual and grasp-specific affordances of tools: an ERP study. *Neuroscience*, 2018, Dec 1;394:243-266. doi: 10.1016/j.neuroscience.2018.10.019.

Gaze study suggesting a mechanism for prioritization of action features driven by likelihood/prioritization of action needs

- Topping KY, Natraj N, Temples D, Atawala N, Gale MK, Wheaton LA. Flexible constraint hierarchy during the visual encoding of tool-object interactions. *European Journal of Neuroscience*, 2021, <https://doi.org/10.1111/ejn.15460>

And a number of fMRI and EEG studies show similar activation profiles as suggested here, with some common implications

-Mizelle, J.C. & Wheaton, L.A. Testing perceptual limits of functional units: are there "automatic" tendencies to associate tools and objects? *Neurosci Lett*; 2011; 488: 92-96.

-Mizelle J.C., Kelly R & Wheaton L.A. Ventral encoding of functional affordances: a neural pathway for identifying errors in action. *Brain and Cognition*, 2013, 82: 274-282.

-Natraj N., Poole V., Mizelle JC, Flumini A., Borghi A, Wheaton L.A. Context and Hand Posture Modulate the Neural Dynamics of Tool-Object Perception. *Neuropsychologia*, 2013, 51: 506-519.

-Mizelle, J.C. & Wheaton, L.A. Why is that hammer in my coffee: A multimodal imaging investigation of contextually-based tool understanding. *Front. Hum. Neurosci*; 2011; 4:233. doi: 10.3389/fnhum.2010.00233

-Schubotz RI, Wurm MF, Wittmann MK, von Cramon DY. Objects tell us what action we can expect:

dissociating brain areas for retrieval and exploitation of action knowledge during action observation in fMRI. *Front Psychol.* 2014 Jun 24;5:636. doi: 10.3389/fpsyg.2014.00636. PMID: 25009519; PMCID: PMC4067566.

-Michalowski B, Buchwald M, Klichowski M, Ras M, Kroliczak G. Action goals and the praxis network: an fMRI study. *Brain Struct Funct.* 2022 Sep;227(7):2261-2284. doi: 10.1007/s00429-022-02520-y. Epub 2022 Jun 22. PMID: 35731447; PMCID: PMC9418102.

As well, consideration of action in social context was recently reviewed in:

-Musco MA, Zazzera E, Paulesu E, Sacheli LM. Error observation as a window on performance monitoring in social contexts? A systematic review. *Neurosci Biobehav Rev.* 2023 Apr;147:105077. doi: 10.1016/j.neubiorev.2023.105077. Epub 2023 Feb 8. PMID: 36758826.

4. Is the work convincing, and if not, what further evidence would be required to strengthen the conclusions?

A better appreciation of the results would be helpful, particularly what is distinct in activation patterns for the different stimuli. This feels a bit buried in the present results presentation, beyond some basic discussion.

It would be good to have a clearer idea of how these tool-object pairs were identified. Was there a norming procedure involved, or were they randomly identified? Is there a difference in perceived complexity which might influence the results (Li X, Krol MA, Jahani S, Boas DA, Tager-Flusberg H, Yücel MA. Brain correlates of motor complexity during observed and executed actions. *Sci Rep.* 2020 Jul 3;10(1):10965. doi: 10.1038/s41598-020-67327-5. PMID: 32620887; PMCID: PMC7335074.)

I am not clear as to what this line means:

"All the objects and object-tool pairs depicted in the stimuli were graphically represented on a table in such a way as to appear closest to the observer, namely in the participant's peri-personal space." (In 458-460) How do you ensure that images appear to be in peripersonal space? Was the presentation of the images somehow 3D also?

Curious about the choice to use computer-generated images versus pictures of real tools-objects in this study. Could this influence realistic perception? What programs were used to generate the images?

In the non-control conditions, there was a motor response with the middle finger of their right hand ("namely their non-dominant arm", which seems to be wrong since all participants were right-handed). But crucially, this was contrasted against a control (Tables 1 & 2, Figure 3). Is it possible that the rapid visuomotor reactions are a part of the (presumed) activity and/or networks seen here?

5. On a more subjective note, do you feel that the paper will influence thinking in the field?

This paper needs some work, but it could shed more light on the connectivity that many researchers have presumed in this field. It would be VERY impactful if better grounded and discussed outside the context of trying to justify their model (which is extensive in the discussion). There are many models of action systems, tool use, etc... which can be described better in other papers. This good empirical paper should focus more on how the results fit in with existing literature in clearer ways.

Reviewer #2 (Remarks to the Author):

In this study, Federico et. al, use neuroimaging (fMRI) connectivity analyses to extend their prior eye tracking study that had shown differing eye movement patterns when evaluating semantic or motor content in images of tool and objects. This led to their framework of 'action reappraisal' and the goal of this current study is to understand the neural correlates of this framework using broadly the same stimuli set of their prior studies. In brief, the authors do find unique neural networks for the type of stimulus and task: with ventral networks more heavily involved in

semantic cognition, and more dorsal contributions relating to understanding the mechanics of the action in the stimuli. Finally, left ventral regions seemed to better distinguish between correct and incorrect tool-object pairs. The paper is well written generally, with a good study design and appropriate analyses and nicely extends the author's prior studies. In my view this study will make a good contribution to the literature. I do have certain general comments and issues, these are outlined below:

Major issues:

Introduction:

- As a neuroimaging paper, there was very little emphasis on the functional neuroanatomy in the Introduction. I encourage the authors to expand more on the functional neuroanatomy part better and focus on it earlier, especially in relation to their hypotheses.

- "Semantic knowledge about tools is neither necessary nor sufficient for individuals to use a hammer on a nail". This statement and the framing in the 2nd and 3rd paragraphs were problematic for me as it is at odds with the authors' later discussion section, their own text and in relation to the field of apraxia for instance, where brain injury causes deficits in exactly the type of functions the authors lay out in the early paragraphs. On one hand, the authors imply that there is a clear disassociation between conceptual and motoric aspects of tool-use based on functional neuroanatomy, and then immediately state in the next paragraph that this is not appropriate and there should be complex interactions. I empathize with the authors with the debate is 'lively' (Ochipa et al., 1989; Goldenberg, 2009; Mizelle and Wheaton, 2010; Mizelle et al., 2013), but if the authors were bring their compelling framework from their prior studies earlier, I feel it would be an easier read. I also felt that the early paragraphs could have more emphasized functional neuroanatomy, as this is the major focus of the paper.

- Lines 105-115 regarding flexible use of tools based on technical reasoning of the goal: I really liked this paragraph. Indeed, the introduction was a pleasure to read once the authors got to their prior study and laid a clear motivation and extension for the neuroimaging hypotheses. However, there have been studies now going on for over two decades on neural responses to tools, to tools when paired with other objects, on the contextual relationships between the tools and objects etc. This part of the introduction would be strengthened if the authors would dive into functional neuroanatomy further (as this is an imaging paper) and outline the similarities and key differences in their neural hypotheses vs. what the neuroimaging field generally posits re: parietofrontal networks.

Results:

- The authors appeared to collect behavioral data but there were no analyses of this. Were the authors earlier findings broadly replicated? And did the authors consider running associative analyses between response times and/or accuracy with neural activations?

Discussion:

- The discussion, in my opinion, could be made shorter and crisper, otherwise as a reader I felt a little lost. There were two paragraphs towards the end for example which start with 'To sum up' and then 'In conclusion'.

- The predominant majority of action perception studies do not consider the role of eye movements on neural activity patterns when processing the stimulus, especially when considering the role of frontal and parietal networks. This is problematic as activity over these areas are heavily influenced by eye movement patterns even when processing similar affordance content (Natraj et al., 2018). Indeed, the authors own prior eye tracking studies have shown differing activity patterns based on the content of stimuli. To what extent are the results therefore influenced by differing gaze strategies? Can the authors expand upon this in the discussion?

1. Goldenberg G (2009) Apraxia and the parietal lobes. *Neuropsychologia* 47:1449-1459.
2. Mizelle J, Wheaton LA (2010) Why is that hammer in my coffee? A multimodal imaging investigation of contextually based tool understanding. *Frontiers in human neuroscience* 4:233.
3. Mizelle J, Kelly RL, Wheaton LA (2013) Ventral encoding of functional affordances: a neural pathway for identifying errors in action. *Brain and cognition* 82:274-282.
4. Natraj N, Alterman B, Basunia S, Wheaton LA (2018) The role of attention and saccades on

parietofrontal encoding of contextual and grasp-specific affordances of tools: An ERP study. *Neuroscience* 394:243-266.

5. Ochipa C, Rothi LG, Heilman KM (1989) Ideational apraxia: A deficit in tool selection and use. *Annals of Neurology: Official Journal of the American Neurological Association and the Child Neurology Society* 25:190-193.

Reviewer #3 (Remarks to the Author):

The presented manuscript describes a single functional MRI study examining the brain networks involved in various components of tool understanding. Specifically, the networks involved in the understanding of tool semantics and the understanding of tool mechanics/physics. The main findings are that a ventral visual stream network was active during the semantic-based task compared to control conditions. When a physical tool mechanics task was performed, a ventrodorsal network was more active than control conditions. The findings are interpreted as support for the action reappraisal theory of tool use. The experiments are well-performed, and the data analysis is appropriate. The conclusions drawn also stem from the results. The manuscript is well-written, although numerous sections have ambiguous terminology. Additionally, many areas of the manuscript would benefit from clarification. The topic is of moderate interest to the readers of *Communications Biology* and, therefore, I would recommend publication provided the following concerns could be addressed.

Major Concerns Which Must Be Addressed:

- 1) Although well-written, there is a fair bit of technical jargon, and the writing style is sometimes awkward throughout the manuscript's introduction and discussion sections. A thorough proofreading with an emphasis on readability is suggested.
- 2) Although the ventral and dorsal corticovisual streams are often referenced, these need to be better introduced and explained early in the manuscript. This may also benefit from a Figure highlighting these systems' differences.
- 3) More detail as to the Frederico and Brandimonte behavioral study should be included. In particular, describing how visuospatial attention was measured and how it was determined to vary from condition to condition.
- 4) Lines 174-175. "...fMRI analysis to the duration of their presentation." This needs further clarification and justification. Since this is a block design, it is entirely possible that task-related responses are being prepared for even though they haven't been explicitly asked to do so at this point in the paradigm.
- 5) Results. Was the data examined for the presence of any order effects? Though randomized, it is still conceivable that task order influenced what the participant was doing during the object presentation epoch.
- 6) Methods/Results. Although the tool-use network is predominantly left-lateralized, many studies show right-hemisphere activity, especially with some tasks performed as part of this experiment. Further justification as to why only the left-hemisphere activity was examined is needed.
- 7) Figure 3. More detail is required in the Figure Legend. What do the t-values represent? The colors? How are the connectivity patterns as displayed interpreted regarding the thickness of lines? Additionally, separating the figure into subsections (A, B, C, etc.) would aid in interpreting this figure.
- 8) More context is required when discussing the anatomical regions, such as the ATL (line 238). The anterior temporal lobe is involved in many processes, and specific segmentation schemas can be variable. It is important to be precise when discussing anatomy in this fashion. Additionally,

when presenting them, acronyms need to be spelled out in all instances (e.g., aITG, aMTG, etc.). Currently, this information only appears in the tables.

9) Discussion. Line 389. The participants are described as having “automatically engaged” different functional networks. The automaticity of this process is ambiguous, as participants knew they would need to make a response as part of the task. From a cognitive processing perspective, whether a task is truly automatic requires a core set of circumstances to be met, which this study has not examined.

10) Methods. Were participants compensated for participating in this study?

11) Methods. Was a power analysis conducted to ensure that 20 participants were adequate for the devised study? Although this is more challenging with fMRI paradigms, it is possible and should be performed (or justification for the lack of an a priori power analysis needs to be provided).

12) Methods. The document describes participants as completing their responses with the index or middle finger of the right non-dominant arm. I believe this is their dominant arm as all participants must be right-handed.

Minor Comments:

a) Abstract , Line 46: Not sure what is meant by “most naturally”

b) Introduction. Line 96 “As regards mechanical..” reword.

c) Introduction. Line 110 “...plastic mechanisms of interpreting phenomenal reality...” not sure what this means, please re-word.

d) Introduction. Line 118 “...reverberate on the temporal allocation of visuospatial attention” reword for clarity.

e) Introduction. Lines 131-132 “...semantic-to-mechanical-to-sensorimotor cascade mechanism...” reword for clarity.

f) Describing the experiment as utilizing “high-field fMRI” is a little misleading. Currently, 1.5 and 3T are considered your “run of the mill” clinical magnets, with 3T+ being considered “high-field.”

g) Line 347 “macro-domain of executive function.” Define/explain what is meant by this.

h) Within the discussion, it needs to be clear that only the “...semantic and mechanical functional brain networks...” within the LEFT hemisphere were examined.

i) Line 422 “...frontal instances...” not sure what this means.

j) The term “ad-hoc” is used throughout (~8 different instances) the document in a somewhat ambiguous way and can likely be removed if the authors mean “when necessary” It provides little additional information to say ad-hoc at each point.

k) Methods. Line 534. “...of the first object-tool stimulus...”. Do you mean just the first stimulus or the first part of the stimulus presentation/trial?

Dear Reviewer(s),

We extend our heartfelt appreciation for your considerate and valuable feedback on our manuscript.

In response to your thoughtful comments, we have worked on substantial revisions throughout the paper, which we have **highlighted** in the new document submitted.

Firstly, we have thoroughly revised the **introduction** and **discussion** sections to better align with the journal's focus and scope. This has resulted in a more comprehensive and biology-oriented narrative framework, **reducing the technical jargon**.

Secondly, we have taken into account all the **references** you recommended, ensuring that our research is firmly rooted in the relevant literature. Moreover, we have expanded on the **anatomical-functional description** of the dorsal, ventro-dorsal, and ventral brain systems, incorporating their neurocognitive, anthropological, and evolutionary biology implications. These enhancements are expected to offer a more comprehensive understanding of the research topic and its broader significance.

In addition, we have included entirely **new images (Figure 1)** to better illustrate our findings and support the revised narrative. These visual aids should contribute to the clarity of the manuscript.

Regarding the **methodological concerns**, we have thoroughly addressed all the issues you raised. Also, we have clarified in the revised version that behavioral data (reaction times) were not recorded, as we used a dummy pad solely for manipulating experimental conditions in this neuroimaging study.

We genuinely value your efforts in evaluating our work, and your expertise has undeniably led to significant improvements in this revision. As a token of our gratitude, we have included an **acknowledgment** of your contributions in the revised manuscript.

Below, you will find a **point-by-point response** to each of your **comments**.

We are optimistic that these revisions will meet your expectations and enhance the quality of the manuscript.

Thank you very much for your time and thoughtful consideration.

Sincerely,

Giovanni Federico
(On behalf of all the authors)

Reviewer #1 (Prof. Lewis Wheaton)

For this review, I will use the recommended review framework provided by the journal in answering 5 core questions. As well, I am referencing several of my papers for consideration. This is not an attempt to collect citations, but an attempt to think about the results in a broader context, which I believe would be helpful.

This manuscript focuses on a very compelling research interest related to the understanding of the neural basis for our understanding of tool usage. This has been a long-standing issue of the field. The study results “considers action understanding as an ability emerging from semantical/conceptual and mechanical/sensorimotor knowledge integration”.

We thank Prof. Wheaton for conducting a thorough review of our work. We have incorporated all the suggested references into the manuscript, expanding the narrative framework beyond the initially proposed scope. We also significantly reduced the action reappraisal framing from this empirical work, moving it towards broader gnoseological perspectives.

I am not very convinced that these conclusions are entirely novel. There has been a longstanding argument that conceptual and sensorimotor knowledge are integrated in action understanding, which is a presumption of many recent imaging studies. It is novel to provide FC analysis as the primary approach, but much of the discussion focuses on “activations” or “co-activations” that are not included in this paper (unless I am missing some other figures).

We simplified the results section and extended the graphical representations of the results with new figures representing all the co-activations indicated in the tables. In the revised manuscript, all and only the co-activations we found are discussed. We thank the reviewer for stressing this.

The authors present some of their work focusing on semantic understanding of tool use, but the authors have not included some science that presents both complimentary and diverging arguments to theirs. These include many papers from my own lab, including:

A behavioral study indicating how action may emerge through visual context

- Borghi, A.M., Flumini, A., Natraj, N., Wheaton, L.A. One hand two objects: emergence of affordance in context. *Brain and Cognition*; 2012; 10: 64-73.

Gaze tracking studies that present a slightly different picture of visual attention driven by action context

- Natraj N, Pella YM, Borghi AM, Wheaton LA. Visual encoding of tool-object affordances. *Neuroscience*, 2015, 310: 512-527.

Gaze + EEG demonstrating how visual attention modulates left and right hemisphere activity during tool use perception

- Natraj N, Alterman BM, Basunia S, Wheaton LA. The role of attention and saccades on parietofrontal encoding of contextual and grasp-specific affordances of tools: an ERP study. *Neuroscience*, 2018, Dec 1;394:243-266. doi: 10.1016/j.neuroscience.2018.10.019.

Gaze study suggesting a mechanism for prioritization of action features driven by likelihood/prioritization of action needs

- Topping KY, Natraj N, Temples D, Atawala N, Gale MK, Wheaton LA. Flexible constraint hierarchy during the visual encoding of tool-object interactions. *European Journal of Neuroscience*, 2021, <https://doi.org/10.1111/ejn.15460>.

And a number of fMRI and EEG studies show similar activation profiles as suggested here, with some common implications:

- Mizelle, J.C. & Wheaton, L.A. Testing perceptual limits of functional units: are there "automatic" tendencies to associate tools and objects? *Neurosci Lett*; 2011; 488: 92-96.

- Mizelle J.C., Kelly R & Wheaton L.A. Ventral encoding of functional affordances: a neural pathway for identifying errors in action. *Brain and Cognition*, 2013, 82: 274–282.

- Natraj N., Poole V., Mizelle JC, Flumini A., Borghi A, Wheaton L.A. Context and Hand Posture Modulate the Neural Dynamics of Tool-Object Perception. *Neuropsychologia*, 2013, 51: 506-519.

- Mizelle, J.C. & Wheaton, L.A. Why is that hammer in my coffee: A multimodal imaging investigation of contextually-based tool understanding. *Front. Hum. Neurosci*; 2011; 4:233. doi: 10.3389/fnhum.2010.00233

- Schubotz RI, Wurm MF, Wittmann MK, von Cramon DY. Objects tell us what action we can expect: dissociating brain areas for retrieval and exploitation of action knowledge during action observation in fMRI. *Front Psychol*. 2014 Jun 24;5:636. doi: 10.3389/fpsyg.2014.00636. PMID: 25009519; PMCID: PMC4067566.

- Michalowski B, Buchwald M, Klichowski M, Ras M, Kroliczak G. Action goals and the praxis network: an fMRI study. *Brain Struct Funct*. 2022 Sep;227(7):2261-2284. doi: 10.1007/s00429-022-02520-y. Epub 2022 Jun 22. PMID: 35731447; PMCID: PMC9418102.

As well, consideration of action in social context was recently reviewed in:

- Musco MA, Zazzera E, Paulesu E, Sacheli LM. Error observation as a window on performance monitoring in social contexts? A systematic review. *Neurosci Biobehav Rev*. 2023 Apr;147:105077. doi: 10.1016/j.neubiorev.2023.105077. Epub 2023 Feb 8. PMID: 36758826.

We have incorporated all the interesting references recommended by the reviewer in the context of the revised manuscript. Also, we substantially modified the narrative framework of our manuscript. For instance, the introduction and discussion have been largely rewritten, and the focus on the action reappraisal has been deleted. We hope the manuscript may appear more balanced regarding the theoretical frameworks discussed. Also, we placed the references suggested by the reviewer in the context of the extensive neurocognitive networks we included in this study. We sincerely appreciate the reviewer for allowing us to expand upon the theoretical framework of our manuscript.

A better appreciation of the results would be helpful, particularly what is distinct in activation patterns for the different stimuli. This feels a bit buried in the present results presentation, beyond some basic discussion.

In the revised manuscript, we simplified and reworded the results for clarity. We also generated new figures that graphically represents the data we reported in tables by using 3D sagittal meshes generated with Surfice (<https://www.nitrc.org/projects/surfire>).

It would be good to have a clearer idea of how these tool-object pairs were identified. Was there a norming procedure involved, or where they randomly identified? Is there a difference in perceived complexity which might influence the results.

Li X, Krol MA, Jahani S, Boas DA, Tager-Flusberg H, Yücel MA. Brain correlates of motor complexity during observed and executed actions. *Sci Rep*. 2020 Jul 3;10(1):10965. doi: 10.1038/s41598-020-67327-5. PMID: 32620887; PMCID: PMC7335074.)

The stimuli utilized in this study were selected based on a jury evaluation. Specifically, we chose stimuli that received homogenous high scores ($M = 4.6$; $SD = 0.52$) in terms of ease of grasp and use. To assess this, we asked a jury comprising 10 individuals (5 females; mean age = 26.4 years; $SD = 2.1$) to rate each tool from the study's stimuli on a Likert scale ranging from 1 (extremely difficult to grasp and use) to 5 (extremely easy to grasp and use). We inserted this information in the methods of the revised manuscript.

I am not clear as to what this line means:

“All the objects and object-tool pairs depicted in the stimuli were graphically represented on a table in such a way as to appear closest to the observer, namely in the participant's peri-personal space.” (In 458-460) How do you ensure that images appear to be in peripersonal space? Was the presentation of the images somehow 3D also?

Information regarding distance and the angle of vision was derived from the estimations provided by the participants, as reported during the post-experimental interview. The stimuli were depicted in an MRI-compatible binocular visor, with a resolution of 1024 x 800 px and a refresh rate of 60Hz. We inserted this information in the methods of the revised manuscript.

Curious about the choice to use computer-generated images versus pictures of real tools-objects in this study. Could this influence realistic perception? What programs were used to generate the images?

Given the level of realism of the stimuli used in the study, we are confident about the fact that their ecological validity should be equal to, if not superior to, the monochromatic and often stylized stimuli used in the literature generally (see Federico & Brandimonte, 2019; Brain & Cognition for discussion on the ecological aspects of the stimuli used in the literature). Accordingly, this study's stimuli received high scores in terms of realism from the jury ($M = 4.8$; $SD = 0.42$). Stimuli were realized with Adobe Photoshop by including vectorial files of the PixelSquid library (<https://www.pixelsquid.com>).

We inserted these details in the methods of the revised manuscript.

In the non-control conditions, there was a motor response with the middle finger of their right hand ("namely their non-dominant arm", which seems to be wrong since all participants were right-handed). But crucially, this was contrasted against a control (Tables 1 & 2, Figure 3). Is it possible that the rapid visuomotor reactions are a part of the (presumed) activity and/or networks seen here?

In the gPPI functional connectivity we focused only on the time window of stimulus presentation, excluding button-pressing responses. If motor responses were present in both the mechanical and semantic conditions, they should produce similar neural outcome as the amount of variance introduced would be comparable. This is not the case of this study's results, which highlights distinct patterns of activations as concerning the dorsal and ventro-dorsal brain regions. In the revised manuscript, we have emphasized these aspects. We extend our gratitude to the reviewer for providing us with the opportunity to present stronger methodological information regarding these important considerations.

This paper needs some work, but it could shed more light on the connectivity that many researchers have presumed in this field. It would be VERY impactful if better grounded and discussed outside the context of trying to justify their model (which is extensive in the discussion). There are many models of action systems, tool use, etc... which can be described better in other papers. This good empirical paper should focus more on how the results fit in with existing literature in clearer ways.

We sincerely appreciate the reviewer's recognition of the potentially significant impact of our results. Of course, as authors of the paper, we acknowledge our subjective position and cannot objectively assess this. However, we are convinced that the theoretical contribution provided by the reviewer has allowed us to fundamentally reshape the argumentative structure of the manuscript.

We have completely deleted in the revised manuscript the narrative highlighting this study as designed to test action reappraisal predictions. Much more space has been given to the interesting complementary literature suggested by the reviewer, as well as further evolutionary biology, cognitive neuroscience and neuroimaging aspects which may find hopefully resonance in a biology journal.

Reviewer #2

In this study, Federico et. al, use neuroimaging (fMRI) connectivity analyses to extend their prior eye tracking study that had shown differing eye movement patterns when evaluating semantic or motor content in images of tool and objects. This led to their framework of 'action reappraisal' and the goal of this current study is to understand the neural correlates of this framework using broadly the same stimuli set of their prior studies. In brief, the authors do find unique neural networks for the type of stimulus and task: with ventral networks more heavily involved in semantic cognition, and more dorsal contributions relating to understanding the mechanics of the action in the stimuli. Finally, left ventral regions seemed to better distinguish between correct and incorrect tool-object pairs. The paper is well written generally, with a good study design and appropriate analyses and nicely extends the author's prior studies. In my view this study will make a good contribution to the literature. I do have certain general comments and issues, these are outlined below:

We thank the reviewer for their diligent and comprehensive review of our work. Their valuable feedback has allowed us to enrich the manuscript by incorporating all the suggested changes and expanding the anatomical-functional narrative of this manuscript.

- As a neuroimaging paper, there was very little emphasis on the functional neuroanatomy in the Introduction. I encourage the authors to expand more on the functional neuroanatomy part better and focus on it earlier, especially in relation to their hypotheses.

In the revised manuscript, the introduction has undergone substantial rewriting. We extensively described all the regions of interest included in this study. This narrative progresses from the dorsal to the ventral regions, encompassing both anatomical-functional and evolutionary biology aspects. Finally, new images (Figure 1) have been included in the manuscript to represent our findings in the context of the brain systems we considered in the functional connectivity analyses.

We thank the reviewer for emphasizing the need of further stressing the neuroanatomical sketch of this manuscript. Indeed, we believe that the revised introduction is now better aligned with the focus of a biology journal like *Comms Biology*.

- "Semantic knowledge about tools is neither necessary nor sufficient for individuals to use a hammer on a nail". This statement and the framing in the 2nd and 3rd paragraphs were problematic for me as it is at odds with the authors' later discussion section, their own text and in relation to the field of apraxia for instance, where brain injury causes deficits in exactly the type of functions the authors lay out in the early paragraphs. On one hand, the authors imply that there is a clear disassociation between conceptual and motoric aspects of tool-use based on functional neuroanatomy, and then immediately state in the next paragraph that this is not appropriate and there should be complex interactions. I empathize with the authors with the debate is 'lively' (Ochipa et al., 1989; Goldenberg, 2009; Mizelle and Wheaton, 2010; Mizelle et al., 2013), but if the authors were bring their compelling framework from their prior studies earlier, I feel it would be an easier read. I also felt that the early paragraphs could have more emphasized functional neuroanatomy, as this is the major focus of the paper.

We agree with the reviewer in considering the initial phrasing of the Introduction a bit confusing concerning semantic knowledge. In the revised manuscript we entirely rewritten these parts, in the context of the new narrative of the Introduction.

- Lines 105-115 regarding flexible use of tools based on technical reasoning of the goal: I really liked this paragraph. Indeed, the introduction was a pleasure to read once the authors got to their prior study and laid a clear motivation and extension for the neuroimaging hypotheses. However, there have been studies now going on for over two decades on neural responses to tools, to tools when paired with other objects, on the contextual relationships between the tools and objects etc. This part of the introduction would be strengthened if the authors would dive into functional neuroanatomy further (as this is an imaging paper) and outline the similarities and key differences in their neural hypotheses vs. what the neuroimaging field generally posits re: parietofrontal networks.

In the revised manuscript, we have significantly expanded the theoretical introduction as well as the anatomical-functional aspects (see responses to previous points).

Results:

- The authors appeared to collect behavioral data but there were no analyses of this. Were the authors earlier findings broadly replicated? And did the authors consider running associative analyses between response times and/or accuracy with neural activations?

In this study, participants' responses were not recorded as we used a dummy (fake) pad, which served only as an instrumental tool for manipulating the experimental conditions.

We stated this in the revised version of the manuscript (methods).

Discussion:

- The discussion, in my opinion, could be made shorter and crisper, otherwise as a reader I felt a little lost. There were two paragraphs towards the end for example which start with 'To sum up' and then 'In conclusion'.

In the revised manuscript, we have made substantial modifications to both the introduction and discussion sections. Considering the suggestions provided by the reviewer, we have removed redundancies and streamlined the discussion. We believe that these changes have resulted in a more coherent and fluent discussion. We thank the reviewer for stressing this.

- The predominant majority of action perception studies do not consider the role of eye movements on neural activity patterns when processing the stimulus, especially when considering the role of frontal and parietal networks. This is problematic as activity over these areas are heavily influenced by eye movement patterns even when processing similar affordance content (Natraj et al., 2018). Indeed, the authors own prior eye tracking studies have shown differing activity patterns based on the content of stimuli. To what extent are the results therefore influenced by differing gaze strategies? Can the authors expand upon this in the discussion?

We expressly included this aspect by underlining in the revised discussion how the interplay of left fronto-parietal regions we found may be influenced not only by high-level effects arising from the interaction between semantic and mechanical knowledge but also by low-level effects (Natraj et al., 2018).

1. Goldenberg G (2009) Apraxia and the parietal lobes. *Neuropsychologia* 47:1449-1459.
2. Mizelle J, Wheaton LA (2010) Why is that hammer in my coffee? A multimodal imaging investigation of contextually based tool understanding. *Frontiers in human neuroscience* 4:233.
3. Mizelle J, Kelly RL, Wheaton LA (2013) Ventral encoding of functional affordances: a neural pathway for identifying errors in action. *Brain and cognition* 82:274-282.
4. Natraj N, Alterman B, Basunia S, Wheaton LA (2018) The role of attention and saccades on parietofrontal encoding of contextual and grasp-specific affordances of tools: An ERP study. *Neuroscience* 394:243-266.
5. Ochipa C, Rothi LG, Heilman KM (1989) Ideational apraxia: A deficit in tool selection and use. *Annals of Neurology: Official Journal of the American Neurological Association and the Child Neurology Society* 25:190-193.

We included all the suggested references in the context of the new narrative structure of the manuscript.

Reviewer #3

The presented manuscript describes a single functional MRI study examining the brain networks involved in various components of tool understanding. Specifically, the networks involved in the understanding of tool semantics and the understanding of tool mechanics/physics. The main findings are that a ventral visual stream network was active during the semantic-based task compared to control conditions. When a physical tool mechanics task was performed, a ventrodorsal network was more active than control conditions. The findings are interpreted as support for the action reappraisal theory of tool use. The experiments are well-performed, and the data analysis is appropriate. The conclusions drawn also stem from the results. The manuscript is well-written, although numerous sections have ambiguous terminology. Additionally, many areas of the manuscript would benefit from clarification. The topic is of moderate interest to the readers of Communications Biology and, therefore, I would recommend publication provided the following concerns could be addressed.

We express our gratitude to the reviewer for their meticulous evaluation of our manuscript.

We have carefully addressed all the suggested changes and have taken the opportunity to further expand the theoretical narrative of our work based on their feedback.

1) Although well-written, there is a fair bit of technical jargon, and the writing style is sometimes awkward throughout the manuscript's introduction and discussion sections. A thorough proofreading with an emphasis on readability is suggested.

Significant modifications have been made to the content of the manuscript.

The introduction and discussion have been extensively reworded, with a particular emphasis on reducing the use of technical jargon, as the reviewer suggested, while expanding the epistemological horizons of the study toward broader theoretical and neuroanatomical implications (see responses to Reviewer 1 and 2). At the same time, we aimed to improve the readability of the manuscript and have taken into account the valuable comments provided by the reviewer. We express our gratitude to the reviewer for their helpful suggestions, which have contributed to enhancing the narrative of our work.

2) Although the ventral and dorsal corticovisual streams are often referenced, these need to be better introduced and explained early in the manuscript. This may also benefit from a Figure highlighting these systems' differences.

In the revised manuscript, we have made significant revisions to the introduction, thoroughly reworking most of its paragraphs. Additionally, we have incorporated a new narrative that provides a comprehensive description of all the regions of interest examined in this study. This narrative follows a progression from the dorsal to the ventral regions, encompassing anatomical-functional and evolutionary biology aspects, which are in line with the aim of the journal. Then, we expressly introduced some behavioral and eye-tracking results from distinct laboratories (see Reviewer 1 comments and responses) and discussed how interactions among dorsal, ventro-dorsal and ventral networks may modulate the temporal allocation of object-related visuospatial attention. Furthermore, we have included new images in the manuscript that visually represent our findings within the context of the brain systems analyzed in the functional connectivity analyses. Specifically, in Figure 1, we have included the functional brain network and ROIs that were considered in the study. This graphical revision was implemented with the goal of providing a clearer and more comprehensive representation of the brain networks investigated in our research.

3) More detail as to the Frederico and Brandimonte behavioral study should be included. In particular, describing how visuospatial attention was measured and how it was determined to vary from condition to condition.

In the revised manuscript, we have made significant changes to the Introduction by expanding the scope beyond the exclusive focus on the eye-tracking data presented in Federico & Brandimonte's studies. The revised manuscript provides a broader narrative that encompasses behavioral studies conducted by multiple laboratories. These studies have documented the direct influence of contextual and sensorimotor knowledge on the temporal allocation of visuospatial attention to different parts of the visual scene. Additionally, we emphasized how complementary evidence from various sources has highlighted the role of semantic and higher-order knowledge in modulating human visual exploration of the environment and the processing of action possibilities. In the revised Introduction, we positioned the results of Federico & Brandimonte within this broader context. This adjustment has modified the narrative structure of the manuscript to demonstrate the

alignment between behavioral studies and neuroimaging findings, which indicate that the human understanding of tool-related actions emerges from interactions among distinct and specialized neurocognitive systems.

4) Lines 174-175. "...fMRI analysis to the duration of their presentation." This needs further clarification and justification. Since this is a block design, it is entirely possible that task-related responses are being prepared for even though they haven't been explicitly asked to do so at this point in the paradigm.

The gPPI analysis focused only on the time window of stimulus presentation, excluding button-pressing responses from the timeseries. If motor responses were present in both the mechanical and semantic conditions, they should produce similar neural outcome as the amount of variance introduced would be comparable. This is not the case of this study's results, which highlights distinct patterns of activations as concerning the dorsal and ventro-dorsal brain regions. In the revised manuscript, we have emphasized these aspects. We thank the reviewer for providing us with the opportunity to present stronger methodological information regarding these important considerations.

5) Results. Was the data examined for the presence of any order effects? Though randomized, it is still conceivable that task order influenced what the participant was doing during the object presentation epoch.

As the reviewer indicated, we randomized the runs and the trials. This should exclude habituation effects. However, we agree with the reviewer concern and expressly test post-hoc order effects by including in CONN the task/block order(s) as a covariate, namely, betas within the GLM. We found the same functional-connectivity patterns, with no significant differences. We thank the reviewer for stressing this.

6) Methods/Results. Although the tool-use network is predominantly left-lateralized, many studies show right-hemisphere activity, especially with some tasks performed as part of this experiment. Further justification as to why only the left-hemisphere activity was examined is needed.

The primary motivation for our study is theoretical, with a specific focus on the mainly left-lateralized nature of the tool-cognition network, as indicated by the most recent literature and stated by the reviewer. Most importantly, while the tool-related semantic knowledge involves more extensively the brain, including predominantly left regions but also some right regions of the brain, the principal and unique neural correlate of mechanical knowledge, namely, technical reasoning, is the left IFG/SMG. Thus, instead of considering a study with a more complex neuroanatomical interpretation (functional connectivity between right and left areas, therefore passing through the corpus callosum), we chose to limit the number of regions to those strictly necessary to demonstrate a functional coupling between left regions *exclusively* belonging to mechanical knowledge and left regions *predominantly* involved in semantic knowledge.

The downside to this approach is that some right ROIs marginally associated with tool-related semantic knowledge have been excluded. However, such potential involvement of right regions would be difficult to interpret in a task-based functional connectivity framework (again, the issue of the corpus callosum). Therefore, this approach allows us to focus on the main aim of the study: demonstrating the interplay between semantic and mechanical neurocognitive systems in tool-related action understanding.

Finally, this approach enabled us to narrow down the number of regions of interest, reducing the required sample size. In fact, with 10 ROIs, the sample size of this study is consistent with other similar studies (Seidel et al., 2023; Cerebral Cortex; Bosch et al., 2023; Exp Brain Research). We have provided detailed descriptions of the involvement of specifically left-lateralized regions throughout the revised manuscript, particularly in the Introduction. Furthermore, we have incorporated a greater level of anatomical-functional details to further highlight the relevance of left-lateralized regions in the context of our study.

7) Figure 3. More detail is required in the Figure Legend. What do the t-values represent? The colors? How are the connectivity patterns as displayed interpreted regarding the thickness of lines? Additionally, separating the figure into subsections (A, B, C, etc.) would aid in interpreting this figure.

We agree with the reviewer, Figure 3 was a mess and we deleted it from this revision.

We extensively modified the graphical contents of the revised manuscript. In particular, Figure 1 has been updated to include several new components: (A) examples of stimuli, (B) the experimental design and visual flow, (C) the functional brain network and regions of interest that were considered in the analyses, and (D-G) the ROI-to-ROI connections as indicated in Tables 1, 2, 3, and 4. These revisions aim to hopefully provide a clearer and more comprehensive representation of the study's concepts and results.

8) More context is required when discussing the anatomical regions, such as the ATL (line 238). The anterior temporal lobe is involved in many processes, and specific segmentation schemas can be variable. It is important to be precise when discussing anatomy in this fashion. Additionally, when presenting them, acronyms need to be spelled out in all instances (e.g., aITG, aMTG, etc.). Currently, this information only appears in the tables.

We agree with the reviewer's feedback. As a result, we extensively rephrased all the sections in the discussion focused on temporal regions associated with object- and tool-related action understanding. Our study primarily centers on the understanding of tool-related actions and, therefore, we have carefully restructured and streamlined the discussion to minimize potential confounding and out-of-scope information regarding a-modal concepts and the anterior temporal lobe. However, we have maintained the overall semantic-cognition framework of the manuscript. We also indicated all the acronyms in the conclusion of the Introduction and Results, just before the discussion. In the discussion, we reduced the number of acronyms for clarity.

9) Discussion. Line 389. The participants are described as having "automatically engaged" different functional networks. The automaticity of this process is ambiguous, as participants knew they would need to make a response as part of the task. From a cognitive processing perspective, whether a task is truly automatic requires a core set of circumstances to be met, which this study has not examined.

We deleted the word "automatically". Thanks for spotting this.

10) Methods. Were participants compensated for participating in this study?

Participants did not receive any financial compensation for their participation in this study. We included this information in the revised manuscript (methods).

11) Methods. Was a power analysis conducted to ensure that 20 participants were adequate for the devised study? Although this is more challenging with fMRI paradigms, it is possible and should be performed (or justification for the lack of an a priori power analysis needs to be provided).

Neurodesign was employed to determine the optimal design and, specifically, neuropower to calculate the sample size for the block design paradigm and the number of regions of interest we considered (10). The analysis resulted in the selection of 18 subjects to achieve an effect (power = 0.8, with Benjamini-Hochberg correction). Importantly, the chosen sample size aligns with most recent and similar studies (Seidel et al., 2023; Cerebral Cortex; Bosch et al., 2023; Experimental Brain Research). We inserted this in the revised methods.

12) Methods. The document describes participants as completing their responses with the index or middle finger of the right non-dominant arm. I believe this is their dominant arm as all participants must be right-handed.

There was a typo in the manuscript. Thank you for bringing this to our attention. In the revised version, we have made the necessary correction to accurately state that the participants used their non-dominant hand, specifically their left hand, to respond.

a) Abstract , Line 46: Not sure what is meant by "most naturally"

We reworded it in "looking at the object-tool pairs with no explicit tasks". We also reworded the abstract substantially for clarity.

b) Introduction. Line 96 "As regards mechanical.." reword.

We substantially edited the manuscript's introduction, and this line has been deleted.

c) Introduction. Line 110 "...plastic mechanisms of interpreting phenomenal reality..." not sure what this means, please re-word.

We reworded it in "flexible interpretations of physical reality".

d) Introduction. Line 118 "...reverberate on the temporal allocation of visuospatial attention" reword for clarity.

We substantially edited the manuscript's introduction, and this line has been deleted.

e) Introduction. Lines 131-132 "...semantic-to-mechanical-to-sensorimotor cascade mechanism..." reword for clarity.

We substantially edited the manuscript's introduction, and this line has been deleted.

f) Describing the experiment as utilizing "high-field fMRI" is a little misleading. Currently, 1.5 and 3T are considered your "run of the mill" clinical magnets, with 3T+ being considered "high-field."

We have removed the term "high-field" from the manuscript in all instances.

g) Line 347 "macro-domain of executive function." Define/explain what is meant by this.

We reworded the paragraph, which now describes the executive functions in terms of cognitive control, information integration, motor timing, *etc.*

h) Within the discussion, it needs to be clear that only the "...semantic and mechanical functional brain networks..." within the LEFT hemisphere were examined.

We modified "brain" in "left brain".

i) Line 422 "...frontal instances..." not sure what this means.

We deleted the words "frontal instances".

j) The term "ad-hoc" is used throughout (~8 different instances) the document in a somewhat ambiguous way and can likely be removed if the authors mean "when necessary" It provides little additional information to say ad-hoc at each point.

We deleted the term "ad-hoc" throughout the manuscript.

k) Methods. Line 534. "...of the first object-tool stimulus...". Do you mean just the first stimulus or the first part of the stimulus presentation/trial?

During each trial of every run, the psychological predictors were modeled based on the presence of the object-tool stimulus on the screen, starting from its onset until its conclusion. We reworded for clarity: "For each trial of each run, the psychological predictors of the FC analysis were modelled by considering the appearance on screen of the object-tool stimulus from its onset to its end."

Reviewers' comments:

Reviewer #1 (Remarks to the Author):

The substantial revisions to this paper are very well done and are helpful for understanding the results better. I have a few minor comments/suggestions:

-there are a high amount of conjunctive adverbs (e.g., however, thus, crucially...) and similar phrases (e.g., At the same time, Intriguingly,...) at the beginning of sentences in this manuscript, which makes it difficult to read occasionally. I would consider reducing these as it is a bit distracting.

-Please clarify whether the jury was distinct from your fMRI participants.

-I am still a bit confused about the ROI selection. I gather that it was based entirely on previous studies of the "action understanding related brain regions" (but not done as a meta-analysis type of approach, which is fine). But it isn't completely clear why the authors didn't focus on a data-driven approach to ROI selection.

-I would cut from "One may wonder...) In 387 to the last paragraph of the discussion as it feels very speculative.

Reviewer #2 (Remarks to the Author):

This is a much improved manuscript and has addressed a lot of comments from the original submission. I commend the authors for diving more into the functional neuroanatomy in this revision. This is more appropriate and makes their study more compelling. There are however a few issues that remain:

Major:

Introduction, second paragraph, Lines 66-81. In my first review I mentioned having issues the way this paragraph is written. If written from the framing of: "how do I generally solve problems in the environment with tools, and how what are the neural correlates" then at a high level it makes sense. However, if it is written from the framing of "how does the brain recognize tools", then it is at conceptual odds with a large body of work, including some of the citations in the paragraph (the Hodges 1999 paper for example). For instance, tools are special as a class of objects as they automatically prime action information over sensorimotor cortex even with just passive viewing, without the need for any problem solving, or recall of any semantic information. The very recognition or observation of tools automatically involves understanding sensorimotor information i.e., action potentiating or action priming effect. This is independent of actually having a problem to solve the tool with or recalling any semantic information of the tool. See for instance all these papers: (Chao & Martin, 2000; Johnson-Frey, 2004; Tucker and Ellis 1998; Grèzes and others 2003; Humphreys and others 2004; James Lewis, 2006, Natraj 2015 etc.). Moreover, lines 130-133 in the authors own words show that tool-use is independent of semantic information. Maybe the issue lies with the terminology? Perhaps the authors use 'semantic' to encompass all that I had written above?

In addition, the authors use semantic to not only mean tool-knowledge but also to mean the simple tool-use recognition task, and the relationship between tool and objects. On second reading, I was thrown off a bit by the multiple definitions of semantic throughout the intro and results section. The definition of mechanical and sensorimotor also seem to overlap quite a bit on the second reading. In the main task, one can very easily swap sensorimotor and mechanical (looking-to-use task). For the most part in the literature, 'mechanical' knowledge is used to refer a tool's action qualities, and not necessarily the physics of the overall task (role of gravity in hanging a picture). Would recommend the authors to edit their text to make the terminology clear.

Methods and results:

As I understand, the authors use a GLM regression framework to evaluate predictability in any brain region from estimated activity in another seed region and given other task-specific predictors plus noise. In this GLM framework, I believe the authors can contrast semantic vs. mechanical

directly or via a second level test of the regression coefficients after the first level (relative to control). Right now, the authors only perform semantic vs. control and mechanical vs. control individually. They then proceed to discuss or qualitatively compare the results of the two tests. However, this does not tell us if the pattern of differences between the two are statistically meaningful (Gelman and Stern, 2006), i.e., the statistical differences between semantic vs. mechanical directly. Similar arguments hold for the comparison between semantically consistent tool-object pairs vs. semantically inconsistent tool-object pairs. In other words, why didn't the authors directly compare the regression weights from the gPPI analyses for semantically consistent vs. inconsistent for specific ROIs. I agree with the first reviewer that focusing on the 'coactivations' or 'functional connectivity' between ROIs during complex tool-use observation is a vitally missing piece of the puzzle relative to earlier papers that have generally shown activity in various nodes. But I feel that the authors can do a better job in the statistics of the comparisons in functional connectivity.

Reviewer #3 (Remarks to the Author):

The authors have done an excellent job of addressing my previous concerns and I am happy to recommend publication at this time.

Dear Reviewer(s),

We sincerely appreciate your second feedback on our manuscript.

In response to your comments, we have worked on several revisions throughout the paper, which we highlighted in the submitted new document.

Below, you will find a **point-by-point response** to each of your **comments**.

Thank you very much for your time and thoughtful consideration.

Sincerely,

Giovanni Federico
(On behalf of all the authors)

Reviewer 1 (Prof. Lewis Wheaton)

The substantial revisions to this paper are very well done and are helpful for understanding the results better. I have a few minor comments/suggestions:

We thank Prof. Wheaton for conducting a second review of our work and for his feedback.

- there are a high amount of conjunctive adverbs (e.g., however, thus, crucially...) and similar phrases (e.g., At the same time, Intriguingly,...) at the beginning of sentences in this manuscript, which makes it difficult to read occasionally. I would consider reducing these as it is a bit distracting.

As suggested by the reviewer, we significantly reduced the use of conjunctive adverbs and similar phrases throughout the revised manuscript.

- Please clarify whether the jury was distinct from your fMRI participants.

We specified that the jury was distinct from fMRI participants (**page 12**).

- I am still a bit confused about the ROI selection. I gather that it was based entirely on previous studies of the "action understanding related brain regions" (but not done as a meta-analysis type of approach, which is fine). But it isn't completely clear why the authors didn't focus on a data-driven approach to ROI selection.

We recognize the value of data-driven approaches when investigating the neural correlates of cognitive processes. However, our study is primarily guided by experimental hypotheses derived from the most recent theoretical literature, as highlighted by the reviewer. Therefore, our approach involved making choices that primarily served theoretical purposes, specifically exploring an interactive model of cognitive functioning in the context of tool-related action understanding. As a result, based on previous research, we identified left-brain neural regions that may be instrumental in highlighting the dynamic interplay between semantic and mechanical knowledge. In essence, the functional connectivity analysis in this study was tailored to serve the theoretical nature of this study. Consequently, the experimental design, sample size, and methods (see the **response to R3, Point 6, Revision 1** below) have been crafted to underscore the theoretical foundation of the work. This emphasis is evident from the title, which commences with "On the functional brain networks."

RESPONSE TO REVIEWER 3, POINT 6, REVISION 1:

6) Methods/Results. Although the tool-use network is predominantly left-lateralized, many studies show right-hemisphere activity, especially with some tasks performed as part of this experiment. Further justification as to why only the left-hemisphere activity was examined is needed.

The primary motivation for our study is theoretical, with a specific focus on the mainly left-lateralized nature of the tool-cognition network, as indicated by the most recent literature and stated by the reviewer. Most importantly, while the tool-related semantic knowledge involves more extensively the brain, including predominantly left regions but also some right regions of the brain, the principal and unique neural correlate of mechanical knowledge, namely, technical reasoning, is the left IFG/SMG. Thus, instead of considering a study with a more complex neuroanatomical interpretation (functional connectivity between right and left areas, therefore passing through the corpus callosum), we chose to limit the number of regions to those strictly necessary to demonstrate a functional coupling between left regions exclusively belonging to mechanical knowledge and left regions predominantly involved in semantic knowledge.

The downside to this approach is that some right ROIs marginally associated with tool-related semantic knowledge have been excluded. However, such potential involvement of right regions would be difficult to interpret in a task-based functional connectivity framework (again, the issue of the corpus callosum). Therefore, this approach allows us to focus on the main aim of the study: demonstrating the interplay between semantic and mechanical neurocognitive systems in tool-related action understanding.

Finally, this approach enabled us to narrow down the number of regions of interest, reducing the required sample size. In fact, with 10 ROIs, the sample size of this study is consistent with other similar studies (Seidel et al., 2023; Cerebral Cortex; Bosch et al., 2023; Exp Brain Research). We have provided detailed descriptions of the involvement of specifically left-lateralized regions throughout the revised manuscript,

particularly in the Introduction. Furthermore, we have incorporated a greater level of anatomical-functional details to further highlight the relevance of left-lateralized regions in the context of our study.

In the revised manuscript, we aimed to emphasize this aspect further. We underlined how our study provides initial support for such an integrated interpretation of action understanding. Subsequent investigations with larger samples and data-driven approaches should further explore these aspects.” This sentence has been added just before the conclusion section (page 11).

We wish to express our sincere appreciation to the reviewer for providing us with another opportunity to enhance the clarity of the epistemological scope of this manuscript.

- I would cut from "One may wonder...) In 387 to the last paragraph of the discussion as it feels very speculative.

We deleted the speculative sentences and slightly modified the conclusion (see also the previous point).

Reviewer #2

This is a much improved manuscript and has addressed a lot of comments from the original submission. I commend the authors for diving more into the functional neuroanatomy in this revision. This is more appropriate and makes their study more compelling. There are however a few issues that remain:

We thank the anonymous reviewer for their second review of our manuscript.

We are pleased that the reviewer found the manuscript much improved and thank them for their further comments to enhance its theoretical impact and results.

- Introduction, second paragraph, Lines 66-81. In my first review I mentioned having issues the way this paragraph is written. If written from the framing of: “how do I generally solve problems in the environment with tools, and how what are the neural correlates” then at a high level it makes sense. However, if it is written from the framing of “how does the brain recognize tools”, then it is at conceptual odds with a large body of work, including some of the citations in the paragraph (the Hodges 1999 paper for example). For instance, tools are special as a class of objects as they automatically prime action information over sensorimotor cortex even with just passive viewing, without the need for any problem solving, or recall of any semantic information. The very recognition or observation of tools automatically involves understanding sensorimotor information i.e., action potentiating or action priming effect. This is independent of actually having a problem to solve the tool with or recalling any semantic information of the tool. See for instance all these papers: (Chao & Martin, 2000; Johnson-Frey, 2004; Tucker and Ellis 1998; Grèzes and others 2003; Humphreys and others 2004; James Lewis, 2006, Natraj 2015 etc.). Moreover, lines 130-133 in the authors own words show that tool-use is independent of semantic information. Maybe the issue lies with the terminology? Perhaps the authors use ‘semantic’ to encompass all that I had written above? In addition, the authors use semantic to not only mean tool-knowledge but also to mean the simple tool-use recognition task, and the relationship between tool and objects. On second reading, I was thrown off a bit by the multiple definitions of semantic throughout the intro and results section. The definition of mechanical and sensorimotor also seem to overlap quite a bit on the second reading. In the main task, one can very easily swap sensorimotor and mechanical (looking-to-use task). For the most part in the literature, ‘mechanical’ knowledge is used to refer a tool’s action qualities, and not necessarily the physics of the overall task (role of gravity in hanging a picture). Would recommend the authors to edit their text to make the terminology clear.

We understand the perplexities of the reviewer and, therefore, in the revised manuscript, we made some modifications in the introduction that underline the “high-level” focus of this fMRI study, which is based on the groundwork laid by some of us in the domain of tool-related action understanding (e.g., Federico and Brandimonte, 2020; *Scientific Reports*) as well as on the differences between sensorimotor and mechanical knowledge (Osiurak & Badets, 2016; Osiurak et al., 2020; Federico et al. 2021; Goldenberg & Spatt, 2009).

As the reviewer correctly pointed out, tools represent a distinct category of objects capable of triggering motor cortex activations upon visual perception, irrespective of their actual utilization (Chao & Martin, 2000; Johnson-Frey, 2004). Of course, we completely agree with such science and included it in the manuscript. Nevertheless,

the activation of motor regions upon seeing a hammer doesn't necessarily imply that the observer intends to use it, recognizes its identity and functions, or appreciates its potential in daily situations (Osiurak & Federico, 2020). Therefore, the "high-level" nature of tool-related action understanding in everyday contexts is at the root of this study. This significantly differs from a study of the neural correlates of merely seeing a tool. Consistent with the high-level nature of the study, we also titled this manuscript "On the functional brain networks involved in tool-related action *understanding*."

By following the reviewer's suggestions, we have now slightly modified the introduction and included a few additional lines to underscore these aspects, hence hopefully providing a clearer definition of the high-level nature of this study, that is, to use the reviewer's words: "How do I generally solve problems in the environment with tools, and how what are the neural correlates." For instance, we underlined how tools represent a distinct category of objects capable of triggering the sensorimotor regions of the human brain upon visual perception, regardless of their immediate utility^{6,7}. Nevertheless, these automatic activations in response to a tool being seen do not necessarily indicate the observer's intention to use it, recognize its identity, or comprehend its potential to solve day-to-day problems⁸⁻¹⁰ (page 3).

We deeply appreciate and thank the reviewer for giving us the opportunity to further contextualize the high-level nature of this study.

- Methods and results: As I understand, the authors use a GLM regression framework to evaluate predictability in any brain region from estimated activity in another seed region and given other task-specific predictors plus noise. In this GLM framework, I believe the authors can contrast semantic vs. mechanical directly or via a second level test of the regression coefficients after the first level (relative to control). Right now, the authors only perform semantic vs. control and mechanical vs. control individually. They then proceed to discuss or qualitatively compare the results of the two tests. However, this does not tell us if the pattern of differences between the two are statistically meaningful (Gelman and Stern, 2006), i.e., the statistical differences between semantic vs. mechanical directly. Similar arguments hold for the comparison between semantically consistent tool-object pairs vs. semantically inconsistent tool-object pairs. In other words, why didn't the authors directly compare the regression weights from the gPPI analyses for semantically consistent vs. inconsistent for specific ROIs. I agree with the first reviewer that focusing on the 'coactivations' or 'functional connectivity' between ROIs during complex tool-use observation is a vitally missing piece of the puzzle relative to earlier papers that have generally shown activity in various nodes. But I feel that the authors can do a better job in the statistics of the comparisons in functional connectivity.

The theoretical arguments provided in response to the reviewer's earlier comments serve as the basis for the methods applied in our study. As the reviewer correctly pointed out, these methods entail incorporating a control condition for comparison with our experimental conditions instead of relying on a parametric comparison between semantic and mechanical conditions. However, such a parametric comparison would not yield meaningful effects, as we anticipate minimal variance within the pooled data. This anticipation is grounded in the understanding that these conditions relate to distinct and dissociable brain networks. Most importantly, this study aims to examine each of these networks rather than compare them.

As stated in the previous point and extensively in the manuscript, we acknowledge the research on the role of sensorimotor knowledge in tool use. However, tool-related semantic and mechanical brain systems have not received equivalent attention. We employed a ROI-to-ROI T-test subtractive functional-connectivity analytical approach to address this gap. This approach incorporated a control condition in which participants were instructed to observe object-tool pairs. The aim was to eliminate the variance attributable to sensorimotor knowledge across all experimental conditions (see **point 1**), thereby individually investigating semantic and mechanical networks. In this approach, "A > B" implies "A – B." Consequently, we removed the shared variance from the two conditions and subsequently discussed the two distinct networks. Also, we excluded sensorimotor regions (ROIs) to maintain consistency with this hypothesis-driven study.

Therefore, while we are sympathetic to the methodological literature mentioned by the reviewer (Gelman and Stern, 2006), we'd like to clarify that our approach does not converge toward summarizing results based on statistical significance, as our findings are discussed, according to the aim of the study, in the presence of a methodologically controlled condition, net of the pooled variance.

To sum up, we proceeded as follows:

1. We postulated the presence of a common source of variance associated with the sensorimotor system activation when looking at tools (Chao & Martin, 2000; *etc.*). To account for this, we included the control condition and excluded sensorimotor ROIs from the FC analyses.
2. We hypothesized the existence of two distinct functional networks: the tool-related semantic network and the mechanical knowledge network (Lesourd et al., 2021; Reynaud et al., 2016; *etc.*). These networks were expected to be inherently different in nature:
 - a. The semantic knowledge network was linked to the semantic condition.
 - b. The mechanical knowledge network was associated with the mechanical condition.

We hope that the additional methodological details, which we included in the revised manuscript, addressed the perplexities raised by the second reviewer. In particular, we specified in a clearer way in the **Methods (page 15)** that FC analyses were devised by positing the existence of a shared source of variance among the experimental conditions—specifically, sensorimotor knowledge. This knowledge is linked to the activation of sensorimotor brain regions when observing tools, regardless of the task demands^{6,7}. Secondly, we hypothesized the existence of two distinct functional networks: the tool-related semantic network and the mechanical knowledge network^{40,137}, represented by the ROIs included in the analyses (see above). Accordingly, differences in FC between experimental conditions were analyzed by contrasting the two experimental tasks with the control task, which acted as the baseline. This approach reduced the variance associated with sensorimotor knowledge across all experimental conditions, thus facilitating the individual investigation of brain networks related to semantic and mechanical knowledge. Also, sensorimotor-related ROIs^{6,7} were excluded from the analyses to maintain consistency with the study's experimental hypotheses.

Of course, future studies may implement larger samples, different experimental designs, and data-driven approaches to investigate specific brain regions associated with both forms of knowledge. In this direction, this study offers an initial indication: the IFG. However, this study's focus does not encompass this aspect. Nonetheless, we have slightly adjusted the manuscript's conclusions to accommodate these potential future investigations and to remind the reader of the high-level nature of the study (see **point 1**). In the conclusions, we underlined that subsequent investigations involving larger samples and data-driven approaches should further explore these aspects and that, at a theoretical level, by speaking on how the brain may deal with tools in everyday-life contexts, this study may foster the debate about how humans may understand and manipulate the environment to devise and use complex tools, techniques, and technologies (**page 11**).

We would like to express our heartfelt gratitude to the reviewer for giving us the opportunity to enhance the clarity of this study's methods through their constructive feedback.

Reviewer #3

The authors have done an excellent job of addressing my previous concerns and I am happy to recommend publication at this time.

We are pleased to acknowledge that all the concerns raised by the reviewer have been thoroughly addressed, and we are grateful for their assessment that the manuscript is now suitable for publication.

Reviewers' comments:

Reviewer #2 (Remarks to the Author):

I thank the authors for thoughtfully considering my reviews and addressing all issues. In general, given the well thought out paper, data, analyses and rebuttals, I recommend their paper for publication, provided they can satisfy my statistical concerns which I outline here again. Previously, there were two main issues points that I had raised earlier. First was the theoretical issue of the high-level conceptualizing. This has been well addressed by the authors and I don't have any issue with that. The second is the issue of statistics. I expand on that a little more in the following paragraph:

Regarding the rebuttal of the authors to my concerns about the statistics, I understand that the authors emphasize this issue. As they state so themselves in the rebuttal, they are only contrasting relative to control based on well grounded hypotheses, but the way the paper has been written, it reads like the experimental conditions have been directly contrasted. This is not the case in reality. The authors should take care in their text wherever appropriate to note that their results indicate only a qualitative and not statistical difference in FC between experimental conditions. Alternatively, the authors can perform this second level statistical test; this should be very straightforward either directly or via carrying the regression weights per subject per ROI to the second level contrast. More concretely, consider the following lines 209 – 210: "Participants recruited distinct brain networks according to the task and semantic consistency". This is only partially true, as short of a direct contrast with each other, we have no way of knowing if each experimental condition recruited a distinct n/w relative to other experimental conditions. Consider in addition as another example the hypotheses that the authors posit in Lines 183: "Observing tool-object pairs in the semantic condition would result in greater coactivation of ventral ROIs while observing tool-object pairs in the mechanical condition would result in larger dorsal and ventro-dorsal co-activations". However, the authors are not performing this direct contrast. Rather, they are contrasting each condition to the control condition. The result of this test does not tell us statistically whether semantic had greater ventral ROIs than mechanical. This same argument lies with the hypothesized differences between semantically consistent and inconsistent pairs; a direct test between them was not performed, and qualitative observations relative to control are interpreted as direct contrast in the text. This is exactly the issue that Gelman and Sterns, 2006 cautioned against. Yes, the ROIs are chosen based on the literature, but are common to all experimental conditions. With respect to the issue of shared variance: direct comparison of (A – B) in most cases is equivalent to (A-C) - (B-C) if the control is constant. Here, this is indeed the case, implying a common covariate for both A and B. This is fairly standard practice. In any case, the authors only perform (A-C) and (B-C) individually and are too liberal in interpreting the results as a contrast of (A-B), when it is not. This issue bleeds into the discussion section also, where it reads as if two conditions have been explicitly contrasted. If the authors can satisfy the editor with some other approach that solves this issue, then that is acceptable as well.

With regard to some specific responses to me, I could not immediately appreciate what they were trying to convey in the following statements:

"However, such a parametric comparison would not yield meaningful effects, as we anticipate minimal variance within the pooled data."

The authors are using a GLM framework for PPI, a parametric test by definition. Why would such a parametric test not work? Are there non-parametric approaches that would work better (such as permutation tests)? And do the authors expect that a direct contrast b/w conditions does not explain a lot of the variance in the data (r^2 for instance?) in the GLM framework?

Reviewer #2 requests:

I thank the authors for thoughtfully considering my reviews and addressing all issues. In general, given the well thought out paper, data, analyses and rebuttals, I recommend their paper for publication, provided they can satisfy my statistical concerns which I outline here again. Previously, there were two main issues points that I had raised earlier. First was the theoretical issue of the high-level conceptualizing. This has been well addressed by the authors and I don't have any issue with that. The second is the issue of statistics. I expand on that a little more in the following paragraph:

Regarding the rebuttal of the authors to my concerns about the statistics, I understand that the authors emphasize this issue. As they state so themselves in the rebuttal, they are only contrasting relative to control based on well grounded hypotheses, but the way the paper has been written, it reads like the experimental conditions have been directly contrasted. This is not the case in reality. The authors should take care in their text wherever appropriate to note that their results indicate only a qualitative and not statistical difference in FC between experimental conditions. Alternatively, the authors can perform this second level statistical test; this should be very straightforward either directly or via carrying the regression weights per subject per ROI to the second level contrast. More concretely, consider the following lines 209 – 210: "Participants recruited distinct brain networks according to the task and semantic consistency". This is only partially true, as short of a direct contrast with each other, we have no way of knowing if each experimental condition recruited a distinct n/w relative to other experimental conditions. Consider in addition as another example the hypotheses that the authors posit in Lines 183: "Observing tool-object pairs in the semantic condition would result in greater coactivation of ventral ROIs while observing tool-object pairs in the mechanical condition would result in larger dorsal and ventro-dorsal co-activations". However, the authors are not performing this direct contrast. Rather, they are contrasting each condition to the control condition. The result of this test does not tell us statistically whether semantic had greater ventral ROIs than mechanical. This same argument lies with the hypothesized differences between semantically consistent and inconsistent pairs; a direct test between them was not performed, and qualitative observations relative to control are interpreted as direct contrast in the text. This is exactly the issue that Gelman and Sterns, 2006 cautioned against. Yes, the ROIs are chosen based on the literature, but are common to all experimental conditions. With respect to the issue of shared variance: direct comparison of $(A - B)$ in most cases is equivalent to $(A - C) - (B - C)$ if the control is constant. Here, this is indeed the case, implying a common covariate for both A and B. This is fairly standard practice. In any case, the authors only perform $(A - C)$ and $(B - C)$ individually and are too liberal in interpreting the results as a contrast of $(A - B)$, when it is not. This issue bleeds into the discussion section also, where it reads as if two conditions have been explicitly contrasted. If the authors can satisfy the editor with some other approach that solves this issue, then that is acceptable as well.

With regard to some specific responses to me, I could not immediately appreciate what they were trying to convey in the following statements: "However, such a parametric comparison would not yield meaningful effects, as we anticipate minimal variance within the pooled data." The authors are using a GLM framework for PPI, a parametric test by definition. Why would such a parametric test not work? Are there non-parametric approaches that would work better (such as permutation tests)? And do the authors expect that a direct contrast b/w conditions does not explain a lot of the variance in the data (r^2 for instance?) in the GLM framework?

Authors response:

We express our gratitude to the reviewer for their feedback. We believe that their comments have enhanced both the narrative clarity and scientific integrity of the manuscript.

As we've stressed previously, and as the reviewer correctly pointed out in their reports, our hypothesis-driven study aims to explore semantic and mechanical functional networks associated with specific tool-related tasks. In experimental terms, our approach involves examining each task individually using a control/baseline condition and then discussing the results *within* those conditions. To put it simply, we seek to understand what happens in the brain when people look at tools for recognition *AND* what occurs when they examine tools for usage. Therefore, our study avoids direct parametric comparisons *between* the conditions, such as assessing whether a particular brain region or network is more or less co-activated in one condition *OR* in the other. This approach guided our study's methods, design, power analyses, narrative, and, most importantly, considerations regarding sample size.

However, upon a careful re-evaluation of the manuscript, we agree with the reviewer in considering that there were instances in the manuscript where our narrative may have inadvertently conveyed the idea of direct comparisons *between* conditions. In this third revision, therefore, we have incorporated all of the reviewer's suggestions, hopefully eliminating any ambiguities from the text. Additionally, we have removed comparative adjectives when discussing brain networks to ensure greater clarity.

We have highlighted all the changes made in the manuscript and provided a *changelog* below in this letter.

We thank again the reviewer for allowing us to fix this important narrative ambiguity.

Sincerely,

Giovanni Federico, PhD
(On behalf of all the authors)

Revision 3 – changelog:

Abstract

Old text: (page 2)

Task-based functional connectivity patterns differed across conditions, with greater co-activations of ventral left-brain regions during the recognition task and larger left dorsal and ventro-dorsal co-activations in the mechanical condition. Also, regardless of the experimental condition, left ventral co-activations distinguished semantically consistent from inconsistent pairs. These findings support most recent accounts highlighting how action understanding might be seen as a product of semantic and mechanical knowledge integration. In addition, these results may help consolidate emerging neural hypotheses that see the inferior parietal and anterior temporal lobes as hubs for the trans-modal integration of physical and conceptual knowledge, respectively.

New text: (page 2)

Task-based left-brain functional connectivity differed within conditions. When compared to the control condition, both the semantic and mechanical conditions exhibited co-activations that included dorsal (precuneus) and ventro-dorsal (inferior frontal gyrus) regions. However, the semantic task was characterized by connectivity patterns involving extensive medial and posterior temporal areas, while the mechanical task recruited inferior parietal and posterior temporal regions. Furthermore, when distinguishing semantically consistent from inconsistent pairs, an anteroposterior frontotemporal neural circuitry was recruited, regardless of the experimental condition. These findings support recent accounts highlighting how action understanding might be seen as an integrated product of semantic and mechanical knowledge. Also, these results may help consolidate emerging hypotheses that see the inferior parietal and anterior temporal lobes as hubs for the trans-modal integration of physical-mechanical and conceptual-semantic knowledge, respectively.

Introduction

Old text: (page 5)

Capitalizing on the research of the last two decades, most recent theoretical models assume tool-related action understanding as the by-product of interactions among semantic, mechanical, and sensorimotor knowledge^{9,19,72,74}. While the contribution of sensorimotor knowledge in tool-related action understanding has been extensively studied in the literature⁷⁵⁻⁷⁹, the functional interactions between tool-related semantic and mechanical brain systems do not enjoy the same level of investigation. Yet, neurocognitive interactions among these systems in individuals' everyday life may give rise to flexible interpretations of physical reality. For instance, individuals may shift the functional knowledge of a screwdriver to a knife if they find themselves unscrewing a screw without having the screwdriver^{80,81}. Such by-analogy problem-solving resolution strategies reflect the interactive nature of tool-related action knowledge. Hence, investigating the interactions between tool-related semantic and mechanical neurocognitive systems might significantly advance our comprehension of physical understanding.

New text: (page 5)

Capitalizing on the research of the last two decades, most recent theoretical models assume tool-related action understanding as an integrated product of semantic, mechanical, and sensorimotor knowledge^{9,19,72,74}. While the contribution of sensorimotor knowledge in tool-related action understanding has been extensively studied in the literature⁷⁵⁻⁷⁹, the functional connectivity of tool-related semantic and mechanical brain systems do not enjoy the same level of investigation. Yet, the combined functioning of these brain systems in individuals' everyday life may give rise to flexible interpretations of physical reality. For instance, individuals may shift the functional knowledge of a screwdriver to a knife if they find themselves unscrewing a screw without having the screwdriver^{80,81}. Such by-analogy problem-solving resolution strategies reflect the polymorphic nature of tool-related action knowledge. Hence, investigating the functional

neural circuitry of tool-related semantic and mechanical neurocognitive systems might significantly advance our comprehension of action understanding.

Old text: (page 5)

This study used functional magnetic resonance imaging (fMRI) to investigate the interactions among dorsal, ventro-dorsal, and ventral brain systems involved in tool-related action understanding.

New text: (page 5)

This study used functional magnetic resonance imaging (fMRI) to explore dorsal, ventro-dorsal, and ventral functional brain systems involved in tool-related action understanding.

Old text: (page 6) → **Suggested by Reviewer 2**

We predicted that functional brain co-activation patterns would differ across the experimental conditions. Specifically, we hypothesized that observing object-tool pairs in the semantic condition would result in greater co-activations of ventral ROIs, while observing the same pairs within the mechanical condition would result in larger dorsal and ventro-dorsal co-activations.

New text: (page 6)

We hypothesized that functional left-brain co-activation patterns would differ within experimental conditions. Specifically, we expected that observing object-tool pairs in the semantic condition – when compared to the control condition – would yield a functional connectivity pattern involving wide ventral/temporal co-activations. On the other hand, when observing these same pairs in the mechanical condition and comparing it to the control condition, we predicted a co-activation pattern mainly involving dorsal and ventro-dorsal ROIs, including the area PF of the left inferior parietal cortex. Lastly, we posited that the temporal neural circuitry would play a role in distinguishing semantically consistent from semantically inconsistent pairs.

Results

Old text: (page 6) → **Suggested by Reviewer 2**

~~Taken together, these results suggest how, during the visual encoding of object-tool pairs, which did not vary with changing experimental conditions, participants recruited distinct brain networks according to the kind of task and the action-related semantic consistency of the pairs.~~

New text: **This sentence has been removed from the revised manuscript.**

Discussion

Old text: (page 7)

The present fMRI study investigated the functional brain networks involved in tool-related action understanding. Participants were presented with images of object-tool pairs that were semantically consistent (e.g., nail-hammer) or inconsistent (e.g., scarf-hammer) for action in three experimental conditions: semantic task (i.e., indicating whether a tool was present in object-tool pairs previously seen), mechanical task (i.e., indicating whether the tool was usable on the object of the pairs), and control condition (i.e., looking at the object-tool pairs with no explicit tasks). Then, we investigated the task-based functional connectivity of specific left-brain ROIs associated with semantic and mechanical knowledge^{17,38–41,46,85}.

New text: (page 7)

The present study used fMRI to explore the functional brain networks involved in tool-related action understanding. Participants were stimulated with computer-generated images depicting object-tool pairs that could be semantically consistent (e.g., notebook-pen) or inconsistent (e.g., bolt-knife) for action. These stimuli were presented in three experimental conditions: the semantic task (where participants determined the presence

of a tool in previously viewed object-tool pairs), the mechanical task (involving the assessment of the tool's usability with the paired object), and the control condition (in which participants simply observed the object-tool pairs without any specific tasks). Subsequently, we investigated task-based functional connectivity of specific left-brain regions associated with tool-related semantic and mechanical knowledge while participants looked at the object-tool pairs, within experimental conditions ^{17,38-41,46,85}. Importantly, we individually inspected the functional connectivity patterns as a function of the task type by comparing each experimental condition with the control condition, which served as the baseline. Also, we limited the analyses to the duration of object-tool pair presentation to prevent task- and motor-related responses that might affect brain co-activations.

Old text:

(page 8)

At the second level of investigation, we aimed to dissociate the neurocognitive functional networks of tool-related mechanical and semantic knowledge. Results showed that the semantic condition produced more ventralized functional co-activations, with a selective engagement of left temporo-medial brain structures (e.g., left pMTG, toMTG, toITG).

New text:

(page 8)

At the second level of investigation, we aimed to explore the functional connectivity patterns involved in tool-related semantic and mechanical knowledge. Results indicated that the semantic condition produced extensive ventralized functional co-activations. Additionally, we observed dorsal and ventro-dorsal co-activations that involved the Precuneus and left IFG. The ventral co-activations were centered around left temporo-medial brain structures, including the left pMTG, toMTG, and toITG.

References

No new references have been added.